# HUMAN-LEVEL PERFORMANCE IN NO-PRESS DIPLOMACY VIA EQUILIBRIUM SEARCH

**Jonathan Gray**,* **Adam Lerer**,* **Anton Bakhtin, Noam Brown**
Facebook AI Research
{jsgray,alerer,yolo,noambrown}@fb.com

## ABSTRACT

Prior AI breakthroughs in complex games have focused on either the purely adversarial or purely cooperative settings. In contrast, Diplomacy is a game of shifting alliances that involves both cooperation and competition. For this reason, Diplomacy has proven to be a formidable research challenge. In this paper we describe an agent for the no-press variant of Diplomacy that combines supervised learning on human data with one-step lookahead search via regret minimization. Regret minimization techniques have been behind previous AI successes in adversarial games, most notably poker, but have not previously been shown to be successful in large-scale games involving cooperation. We show that our agent greatly exceeds the performance of past no-press Diplomacy bots, is unexploitable by expert humans, and ranks in the top 2% of human players when playing anonymous games on a popular Diplomacy website.

## 1 INTRODUCTION

A primary goal for AI research is to develop agents that can act optimally in real-world multi-agent interactions (i.e., **games**). In recent years, AI agents have achieved expert-level or even superhuman performance in benchmark games such as backgammon (Tesauro, 1994), chess (Campbell et al., 2002), Go (Silver et al., 2016; 2017; 2018), poker (Moravčík et al., 2017; Brown & Sandholm, 2017; 2019b), and real-time strategy games (Berner et al., 2019; Vinyals et al., 2019). However, previous large-scale game AI results have focused on either purely competitive or purely cooperative settings. In contrast, real-world games, such as business negotiations, politics, and traffic navigation, involve a far more complex mixture of cooperation and competition. In such settings, the theoretical grounding for the techniques used in previous AI breakthroughs falls apart.

In this paper we augment neural policies trained through imitation learning with regret minimization search techniques, and evaluate on the benchmark game of no-press Diplomacy. Diplomacy is a longstanding benchmark for research that features a rich mixture of cooperation and competition. Like previous researchers, we evaluate on the widely played no-press variant of Diplomacy, in which communication can only occur through the actions in the game (i.e., no cheap talk is allowed).

Specifically, we begin with a **blueprint** policy that approximates human play in a dataset of Diplomacy games. We then improve upon the blueprint during play by approximating an equilibrium for the current phase of the game, assuming all players (including our agent) play the blueprint for the remainder of the game. Our agent then plays its part of the computed equilibrium. The equilibrium is computed via regret matching (RM) (Blackwell et al., 1956; Hart & Mas-Colell, 2000).

Search via RM has led to remarkable success in poker. However, RM only converges to a Nash equilibrium in two-player zero-sum games and other special cases, and RM was never previously shown to produce strong policies in a mixed cooperative/competitive game as complex as no-press Diplomacy. Nevertheless, we show that our agent exceeds the performance of prior agents and for the first time convincingly achieves human-level performance in no-press Diplomacy. Specifically, we show that our agent soundly defeats previous agents, that our agent is far less exploitable than previous agents, that an expert human cannot exploit our agent even in repeated play, and, most

---

*Equal Contribution.

importantly, that our agent ranks in the top 2% of human players when playing anonymous games on a popular Diplomacy website.

## 2   BACKGROUND AND RELATED WORK

Search has previously been used in almost every major game AI breakthrough, including backgammon (Tesauro, 1994), chess (Campbell et al., 2002), Go (Silver et al., 2016; 2017; 2018), poker (Moravčík et al., 2017; Brown & Sandholm, 2017; 2019b), and Hanabi (Lerer et al., 2020). A major exception is real-time strategy games (Vinyals et al., 2019; Berner et al., 2019). Similar to SPARTA as used in Hanabi (Lerer et al., 2020), our agent conducts one-ply lookahead search (i.e., changes the policy just for the current game turn) and thereafter assumes all players play according to the blueprint. Similar to the Pluribus poker agent (Brown & Sandholm, 2019b), our search technique uses regret matching to compute an approximate equilibrium. In a manner similar to the sampled best response algorithm of Anthony et al. (2020), we sample a limited number of actions from the blueprint policy rather than search over all possible actions, which would be intractable.

Learning effective policies in games involving cooperation and competition has been studied extensively in the field of multi-agent reinforcement learning (MARL) (Shoham et al., 2003). Nash-Q and CE-Q applied Q learning for general sum games by using Q values derived by computing Nash (or correlated) equilibrium values at the target states (Hu & Wellman, 2003; Greenwald et al., 2003). Friend-or-foe Q learning treats other agents as either cooperative or adversarial, where the Nash Q values are well defined Littman (2001). The recent focus on "deep" MARL has led to learning rules from game theory such as fictitious play and regret minimization being adapted to deep reinforcement learning (Heinrich & Silver, 2016; Brown et al., 2019), as well as work on game-theoretic challenges of mixed cooperative/competitive settings such as social dilemmas and multiple equilibria in the MARL setting (Leibo et al., 2017; Lerer & Peysakhovich, 2017; 2019).

Diplomacy in particular has served for decades as a benchmark for multi-agent AI research (Kraus & Lehmann, 1988; Kraus et al., 1994; Kraus & Lehmann, 1995; Johansson & Håård, 2005; Ferreira et al., 2015). Recently, Paquette et al. (2019) applied imitation learning (IL) via deep neural networks on a dataset of more than 150,000 Diplomacy games. This work greatly improved the state of the art for no-press Diplomacy, which was previously a handcrafted agent (van Hal, 2013). Paquette et al. (2019) also tested reinforcement learning (RL) in no-press Diplomacy via Advantage Actor-Critic (A2C) (Mnih et al., 2016). Anthony et al. (2020) introduced sampled best response policy iteration, a self-play technique, which further improved upon the performance of Paquette et al. (2019).

### 2.1   DESCRIPTION OF DIPLOMACY

The rules of no-press Diplomacy are complex; a full description is provided by Paquette et al. (2019). No-press Diplomacy is a seven-player zero-sum board game in which a map of Europe is divided into 75 provinces. 34 of these provinces contain supply centers (SCs), and the goal of the game is for a player to control a majority (18) of the SCs. Each players begins the game controlling three or four SCs and an equal number of units.

The game consists of three types of phases: movement phases in which each player assigns an order to each unit they control, retreat phases in which defeated units retreat to a neighboring province, and adjustment phases in which new units are built or existing units are destroyed.

During a movement phase, a player assigns an order to each unit they control. A unit's order may be to hold (defend its province), move to a neighboring province, convoy a unit over water, or support a neighboring unit's hold or move order. Support may be provided to units of any player. We refer to a tuple of orders, one order for each of a player's units, as an **action**. That is, each player chooses one action each turn. There are an average of 26 valid orders for each unit (Paquette et al., 2019), so the game's branching factor is massive and on some turns enumerating all actions is intractable.

Importantly, all actions occur simultaneously. In live games, players write down their orders and then reveal them at the same time. This makes Diplomacy an imperfect-information game in which an optimal policy may need to be stochastic in order to prevent predictability.

Diplomacy is designed in such a way that cooperation with other players is almost essential in order to achieve victory, even though only one player can ultimately win.

A game may end in a draw on any turn if all remaining players agree. Draws are a common outcome among experienced players because players will often coordinate to prevent any individual from reaching 18 centers. The two most common scoring systems for draws are **draw-size scoring (DSS)**, in which all surviving players equally split a win, and **sum-of-squares scoring (SoS)**, in which player $i$ receives a score of $\frac{C_i^2}{\sum_{j \in \mathcal{N}} C_j^2}$, where $C_i$ is the number of SCs that player $i$ controls (Fogel, 2020). Throughout this paper we use SoS scoring except in anonymous games against humans where the human host chooses a scoring system.

## 2.2 REGRET MATCHING

**Regret Matching (RM)** (Blackwell et al., 1956; Hart & Mas-Colell, 2000) is an iterative algorithm that converges to a Nash equilibrium (NE) (Nash, 1951) in two-player zero-sum games and other special cases, and converges to a coarse correlated equilibrium (CCE) (Hannan, 1957) in general.

We consider a game with $\mathcal{N}$ players where each player $i$ chooses an action $a_i$ from a set of actions $\mathcal{A}_i$. We denote the joint action as $a = (a_1, a_2, \ldots, a_N)$, the actions of all players other than $i$ as $a_{-i}$, and the set of joint actions as $\mathcal{A}$. After all players simultaneously choose an action, player $i$ receives a reward of $v_i(a)$ (which can also be represented as $v_i(a_i, a_{-i})$). Players may also choose a probability distribution over actions, where the probability of action $a_i$ is denoted $\pi_i(a_i)$ and the vector of probabilities is denoted $\pi_i$.

Normally, each iteration of RM has a computational complexity of $\Pi_{i \in \mathcal{N}} |\mathcal{A}_i|$. In a seven-player game, this is typically intractable. We therefore use a sampled form of RM in which each iteration has a computational complexity of $\sum_{i \in \mathcal{N}} |\mathcal{A}_i|$. We now describe this sampled form of RM.

Each agent $i$ maintains an **external regret** value for each action $a_i \in \mathcal{A}_i$, which we refer to simply as **regret**. The regret on iteration $t$ is denoted $R_i^t(a_i)$. Initially, all regrets are zero. On each iteration $t$ of RM, $\pi_i^t(a_i)$ is set according to

$$\pi_i^t(a_i) = \begin{cases} \frac{\max\{0, R_i^t(a_i)\}}{\sum_{a_i' \in \mathcal{A}_i} \max\{0, R_i^t(a_i')\}} & \text{if } \sum_{a_i' \in \mathcal{A}_i} \max\{0, R_i^t(a_i')\} > 0 \\ \frac{1}{|\mathcal{A}_i|} & \text{otherwise} \end{cases} \tag{1}$$

Next, each player samples an action $a_i^*$ from $\mathcal{A}_i$ according to $\pi_i^t$ and all regrets are updated such that

$$R_i^{t+1}(a_i) = R_i^t(a_i) + v_i(a_i, a_{-i}^*) - \sum_{a_i' \in \mathcal{A}_i} \pi_i^t(a_i') v_i(a_i', a_{-i}^*) \tag{2}$$

This sampled form of RM guarantees that $R_i^t(a_i) \in \mathcal{O}(\sqrt{t})$ with high probability (Lanctot et al., 2009). If $R_i^t(a_i)$ grows sublinearly for all players' actions, as is the case in RM, then the *average* policy over all iterations converges to a NE in two-player zero-sum games and in general the empirical distribution of players' joint policies converges to a CCE as $t \to \infty$.

In order to improve empirical performance, we use linear RM (Brown & Sandholm, 2019a), which weighs updates on iteration $t$ by $t$.[1] We also use optimism (Syrgkanis et al., 2015), in which the most recent iteration is counted twice when computing regret. Additionally, the action our agent ultimately plays is sampled from the *final* iteration's policy, rather than the average policy over all iterations. This reduces the risk of sampling a non-equilibrium action due to insufficient convergence. We explain this modification in more detail in Appendix F.

## 3 AGENT DESCRIPTION

Our agent is composed of two major components. The first is a **blueprint** policy and state-value function trained via imitation learning on human data. The second is a search algorithm that utilizes the blueprint. This algorithm is executed on every turn, and approximates an equilibrium policy (for all players, not just the agent) for the current turn via RM, assuming that the blueprint is played by all players for the remaining game beyond the current turn.

---

[1]In practice, rather than weigh iteration $t$'s updates by $t$ we instead discount prior iterations by $\frac{t}{t+1}$ in order to reduce numerical instability. The two options are mathematically equivalent.

### 3.1 SUPERVISED LEARNING

We construct a blueprint policy via imitation learning on a corpus of 46,148 Diplomacy games collected from online play, building on the methodology and model architecture described by Paquette et al. (2019) and Anthony et al. (2020). A blueprint policy and value function estimated from human play is ideal for performing search in a general-sum game, because it is likely to realistically approximate state values and other players' actions when playing with humans. Our blueprint supervised model is based on the DipNet agent from Paquette et al. (2019), but we make a number of modifications to the architecture and training. A detailed description of the architecture is provided in Appendix A; in this section we highlight the key differences from prior work.

We trained the blueprint policy using only a subset of the data used by Paquette et al. (2019), specifically those games obtained from `webdiplomacy.net`. For this subset of the data, we obtained metadata about the press variant (full-press vs. no-press) which we add as a feature to the model, and anonymized player IDs for the participants in each game. Using the IDs, we computed ratings $s_i$ for each player $i$ and only trained the policy on actions from players with above-average ratings. Appendix B describes our method for computing these ratings.

Our model closely follows the architecture of Paquette et al. (2019), with additional dropout of 0.4 between GNN encoder layers. We model sets of build orders as single tokens because there are a small number of build order combinations and it is tricky to predict *sets* auto-regressively with teacher forcing. We adopt the encoder changes of Anthony et al. (2020), but do not adopt their relational order decoder because it is more expensive to compute and leads to only marginal accuracy improvements after tuning dropout.

We make a small modification to the encoder GNN architecture that improves modeling. In addition to the standard residual that skips the entire GNN layer, we replace the graph convolution[2] with the sum of a graph convolution and a linear layer. This allows the model to learn a hierarchy of features for each graph node (through the linear layer) without requiring a concomitant increase in graph smoothing (the GraphConv). The resulting GNN layer computes (modification in red)

$$x_{i+1} = Dropout(ReLU(BN(GraphConv(x_i) + \mathbf{Ax_i}))) + x_i. \qquad (3)$$

where $A$ is a learned linear transformation.

Finally, we achieve a substantial improvement in order prediction accuracy using a *featurized order decoder*. Diplomacy has over 13,000 possible orders, many of which will be observed infrequently in the training data. Therefore, by featurizing the orders by the order type, and encodings of the source, destination, and support locations, we observe improved prediction accuracy.

Specifically, in a standard decoder each order $o$ has a learned representation $e_o$, and for some board encoding $x$ and learned order embedding $e_o$, $P(o) = softmax(x \cdot e_o)$. With order featurization, we use $\tilde{e}_o = e_o + Af_o$, where $f_o$ are static order features and $A$ is a learned linear transformation. The order featurization we use is the concatenation of the one-hot order type with the board encodings for the source, destination, and support locations. We found that representing order location features by their location encodings works better than one-hot locations, presumably because the model can learn more state-contextual features.[3]

We add an additional value head to the model immediately after the dipnet encoder, that is trained to estimate the final SoS scores given a board situation. We use this value estimate during equilibrium search (Sec. 3.2) to estimate the value of a Monte Carlo rollout after a fixed number of steps. The value head is an MLP with one hidden layer that takes as input the concatenated vector of all board position encodings. A softmax over powers' SoS scores is applied at the end to enforce that all players' SoS scores sum to 1.

---

[2]As noted by Anthony et al. (2020), DipNet uses a variant of a GNN that learns a separate weight matrix at each graph location.

[3]We note the likelihood that a transformer with token-based decoding should capture a similar featurization, although Paquette et al. (2019) report worse performance for both a transformer and token-based decoding.

[4]This is slightly different than reported in Paquette et al. (2019) because we compute token-accuracy treating a full set of build orders as a single token.

[5]Policy accuracy for these model are not comparable to above because training data was modified.

| Model | Policy Accuracy | SoS v. DipNet | |
| --- | --- | --- | --- |
| | | *temp=0.5* | *temp=0.1* |
| DipNet (Paquette et al. (2019)) | 60.5%[4] | 0.143 | |
| + combined build orders & encoder dropout | 62.0% | | |
| + encoder changes from Anthony et al. (2020) | 62.4% | 0.150 | 0.198 |
| *switch to webdiplomacy training data only* | 61.3%[5] | 0.175 | 0.206 |
| + output featurization | 62.0% | 0.184 | 0.188 |
| + improved GNN layer | 62.4% | 0.183 | 0.205 |
| + merged GNN trunk | 62.9% | 0.199 | 0.202 |

Table 1: Effect of model and training data changes on supervised model quality. We measure policy accuracy as well as average SoS score achieved by each agent against 6 of the original DipNet model. We measure the SoS scores in two settings: with all 7 agents sampling orders at a temperature of either 0.5 or 0.1.

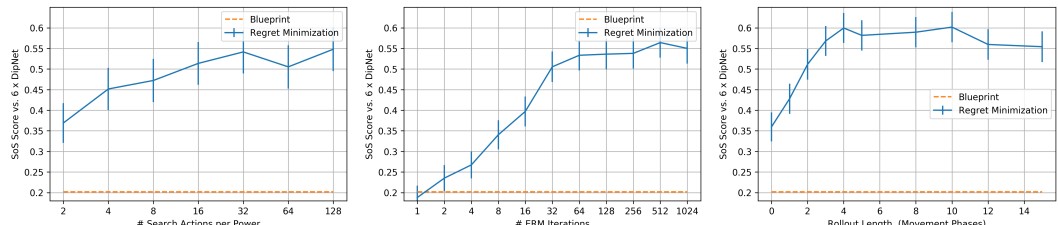

Figure 1: **Left:** Score of SearchBot using different numbers of sampled subgame actions, against 6 DipNet agents ((Paquette et al., 2019) at temperature 0.1). A score of 14.3% would be a tie. Even when sampling only two actions, SearchBot dramatically outperforms our blueprint, which achieves a score of 20.2%. **Middle:** The effect of the number iterations of sampled regret matching on SearchBot performance. **Right:** The effect of different rollout lengths on SearchBot performance.

## 3.2 EQUILIBRIUM SEARCH

The policy that is actually played results from a search algorithm which utilizes the blueprint policy. Let $s$ be the current state of the game. On each turn, the search algorithm computes an equilibrium for a **subgame** and our agent plays according to its part of the equilibrium solution for its next action.

Conceptually, the subgame is a well-defined game that begins at state $s$. The set of actions available to each player is a subset of the possible actions in state $s$ in the full game, and are referred to as the **subgame actions**. Each player $i$ chooses a subgame action $a_i$, resulting in joint subgame action $a$. After $a$ is taken, the players make no further decisions in the subgame. Instead, the players receive a reward corresponding to the players sampling actions according to the blueprint policy $\pi^b$ for the remaining game.

The subgame actions for player $i$ are the $Mk_i$ highest-probability actions according to the blueprint model, where $k_i$ is the number of units controlled by player $i$ and $M$ is a hyperparameter (usually set to 5). The effect of different numbers of subgame actions is plotted in Figure 1 (left).

Rolling out $\pi^b$ to the end of the game is very expensive, so in practice we instead roll out $\pi^b$ for a small number of turns (usually 2 or 3 movement phases in our experiments) until state $s'$ is reached, and then use the value for $s'$ from the blueprint's value network as the reward vector. Figure 1 (right) shows the performance of our search agent using different rollout lengths. We do not observe improved performance for rolling out farther than 3 or 4 movement phases.

We compute a policy for each agent by running the sampled regret matching algorithm described in Section 2.2. The search algorithm used $256 - 4{,}096$ iterations of RM and typically required between 2 minutes and 20 minutes per turn using a single Volta GPU and 8 CPU cores, depending on the hyperparameters used for the game. Details on the hyperparameters we used are provided in Appendix G.

## 4 RESULTS

Using the techniques described in Section 3, we developed an agent we call SearchBot. Our experiments focus on two formats. The first evaluates the head-to-head performance of SearchBot playing against the population of human players on a popular Diplomacy website, as well as against prior AI agents. The second measures the exploitability of SearchBot.

### 4.1 PERFORMANCE AGAINST A POPULATION OF HUMAN PLAYERS

The ultimate test of an AI system is how well it performs in the real world with humans. To measure this, we had SearchBot anonymously play no-press Diplomacy games on the popular Diplomacy website `webdiplomacy.net`. Since there are 7 players in each game, average human performance is a score of 14.3%. In contrast, SearchBot scored 26.6% $\pm$ 3.2%.[6] Table 2 shows the agent's performance and a detailed breakdown is presented in Table 6 in Appendix G. A qualitative analysis of the bot's play by three-time world Diplomacy champion Andrew Goff is presented in Appendix I.

| Power | Bot Score | Human Mean | Games | Wins | Draws | Losses |
|---|---|---|---|---|---|---|
| All Games | 26.6% $\pm$ 3.2% | 14.3% | 116 | 16 | 43 | 57 |

Table 2: Average SoS score of our agent in anonymous games against humans on `webdiplomacy.net`. Average human performance is 14.3%. Score in the case of draws was determined by the rules of the joined game. The $\pm$ shows one standard error.

In addition to raw score, we measured SearchBot's performance using the Ghost-Rating system (Anthony, 2020), which is a Diplomacy rating system inspired by the Elo system that accounts for the relative strength of opponents and that is used to semi-officially rank players on `webdiplomacy.net`. Among no-press Diplomacy players on the site, our agent ranked 17 out of 901 players with a Ghost-Rating of 183.4.[7] Details on the setup for experiments are provided in Appendix G.

### 4.2 PERFORMANCE AGAINST OTHER AI AGENTS

In addition to playing against humans, we also evaluated SearchBot against prior AI agents from Paquette et al. (2019) (one based on supervised learning, and another based on reinforcement learning), our blueprint agent, and a best-response agent (BRBot). Table 3 shows these results. Following prior work, we compute average scores in '1v6' games containing a single agent of type A and six agents of type B. The average score of an identical agent should therefore be $1/7 \approx 14.3\%$. SearchBot achieves its highest 1v6 score when matched against its own blueprint, since it is most accurately able to approximate the behavior of that agent. It outperforms all three agents by a large margin, and none of the three baselines is able to achieve a score of more than $1\%$ against our search agent.

BRBot is an agent that computes a best response to the blueprint policy in the subgame. Like SearchBot, BRBot considers the $Mk_i$ highest-probability actions according to the blueprint. Unlike SearchBot, BRBot searches over only its own actions, and rather than computing an equilibrium, it plays the action that yields the greatest reward assuming other powers play the blueprint policy.

Finally, we ran 25 games between one SearchBot and six copies of Albert (van Hal, 2013), the state-of-the-art rule-based bot. SearchBot achieved an average SoS score of 69.3%.

### 4.3 EXPLOITABILITY

While performance of an agent within a population of human players is the most important metric, that metric alone does not capture how the population of players might adapt to the agent's presence.

---

[6]At the start of the game, the bot is randomly assigned to one of 7 powers in the game. Some powers tend to perform better than others, so the random assignment may influence the bot's score. To control for this, we also report the bot's performance when each of the 7 powers is weighed equally. Its score in this case increases to 26.9% $\pm$ 3.3%. Most games used draw-size scoring in case of draws, while our agent was trained based on sum-of-squares scoring. If all games are measured using sum-of-squares, our agent scores 30.9% $\pm$ 3.5%.

[7]Our agent played games under different accounts; we report the Ghost-Rating for these accounts merged.

| 1x ↓ 6x → | DipNet | DipNet RL | Blueprint | BRBot | SearchBot |
|---|---|---|---|---|---|
| DipNet | - | $6.7\% \pm 0.9\%$ | $11.6\% \pm 0.1\%$ | $0.1\% \pm 0.1\%$ | $0.7\% \pm 0.2\%$ |
| DipNet RL | $18.9\% \pm 1.4\%$ | - | $10.5\% \pm 1.1\%$ | $0.1\% \pm 0.1\%$ | $0.6\% \pm 0.2\%$ |
| Blueprint | $20.2\% \pm 1.3\%$ | $7.5\% \pm 1.0\%$ | - | $0.3\% \pm 0.1\%$ | $0.9\% \pm 0.2\%$ |
| BRBot | $67.3\% \pm 1.0\%$ | $43.7\% \pm 1.0\%$ | $69.3\% \pm 1.7\%$ | - | $11.1\% \pm 1.1\%$ |
| SearchBot | $51.1\% \pm 1.9\%$ | $35.2\% \pm 1.8\%$ | $52.7\% \pm 1.3\%$ | $17.2\% \pm 1.3\%$ | - |

Table 3: Average SoS scores achieved by each of the agents against each other. DipNet agents from (Paquette et al., 2019) and the Blueprint agent use a temperature of 0.1. BRBot scores higher than SearchBot against the Blueprint, but SearchBot outperforms BRBot in a head-to-head comparison.

For example, if our agent is extremely strong then over time other players might adopt the bot's playstyle. As the percentage of players playing like the bot increases, other players might adopt a policy that seeks to exploit this playstyle. Thus, if the bot's policy is highly exploitable then it might eventually do poorly even if it initially performs well against the population of human players.

Motivated by this, we measure the **exploitability** of our agent. Exploitability of a policy profile $\pi$ (denoted $e(\pi)$) measures worst-case performance when all but one agents follows $\pi$. Formally, the exploitability of $\pi$ is defined as $e(\pi) = \sum_{i \in \mathcal{N}} (\max_{\pi_i^*} v_i(\pi_i^*, \pi_{-i}) - v_i(\pi))/N$, where $\pi_{-i}$ denotes the policies of all players other than $i$. Agent $i$'s **best response** to $\pi_{-i}$ is defined as $BR(\pi_{-i}) = \arg\max_{\pi_i} v_i(\pi_i, \pi_{-i})$.

We estimate our agent's full-game exploitability in two ways: by training an RL agent to best respond to the bot, and by having expert humans repeatedly play against six copies of the bot. We also measure the 'local' exploitability in the search subgame and show that it converges to an approximate Nash equilibrium.

### 4.3.1 PERFORMANCE AGAINST A BEST-RESPONDING AGENT

When the policies of all players but one are fixed, the game becomes a Markov Decision Process (MDP) (Howard, 1960) for the non-fixed player because the actions of the fixed players can be viewed as stochastic transitions in the "environment". Thus, we can estimate the exploitability of $\pi$ by first training a best response policy $BR(\pi_{-i})$ for each agent $i$ using any single-agent RL algorithm, and then computing $\sum_{i \in \mathcal{N}} v_i(BR(\pi_{-i}), \pi_{-i})/N$. Since the best response RL policy will not be an exact best response (which is intractable to compute in a game as complex as no-press Diplomacy) this only gives us a lower-bound estimate of the exploitability.

Following other work on environments with huge action spaces (Vinyals et al., 2019; Berner et al., 2019), we use a distributed asynchronous actor-critic RL approach to optimize the exploiter policy (Espeholt et al., 2018). We use the same architecture for the exploiter agent as for the fixed model. Moreover, to simplify the training we initialize the exploiter agent from the fixed model.

We found that training becomes unstable when the policy entropy gets too low. The standard remedy is to use an entropy regularization term. However, due to the immense action space, an exact computation of the entropy term, $E_a \log p_\theta(a)$, is infeasible. Instead, we optimize a surrogate loss that gives an unbiased estimate of the gradient of the entropy loss (see Appendix D). We found this to be critical for the stability of the training.

Training an RL agent to exploit SearchBot is prohibitively expensive. Even when choosing hyperparameters that would result in the agent playing as fast as possible, SearchBot typically requires at least a full minute in order to act each turn. Instead, we collect a dataset of self-play games of SearchBot and train a supervised agent on this dataset. The resulting agent, which we refer to as SearchBot-clone, is weaker than SearchBot but requires only a single pass through the neural network in order to act on a turn. By training an agent to exploit SearchBot-clone, we can obtain a (likely) upper bound on what the performance would be if a similar RL agent were trained against SearchBot.

We report the reward of the exploiter agents against the blueprint and SearchBot-clone agents in Figure 2. The results show that SearchBot-clone is highly exploitable, with a best responder able to score at least 42% against SearchBot-clone. Any score above 14.3% means the best responder is winning in expectation. However, SearchBot-clone appears to be much less exploitable than the

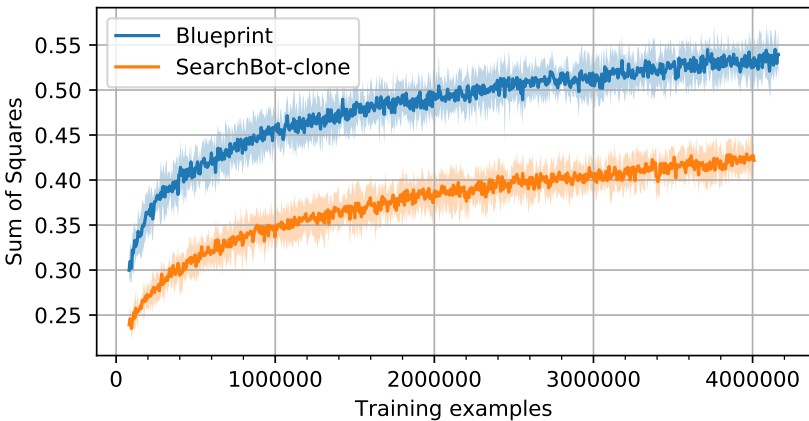

Figure 2: Score of the exploiting agent against the blueprint and SearchBot-clone as a function of training time. We report the average of six runs. The shaded area corresponds to three standard errors. We use temperature 0.5 for both agents as it minimizes exploitability for the blueprint. Since SearchBot-clone is trained through imitation learning of SearchBot, *the exploitability of SearchBot is almost certainly lower than SearchBot-clone.*

blueprint, which a best responder can beat with a score of at least 54%. Also, we emphasize again that SearchBot is almost certainly far less exploitable than SearchBot-clone.

### 4.3.2 PERFORMANCE AGAINST EXPERT HUMAN EXPLOITERS

In addition to training a best-responding agent, we also invited Doug Moore and Marvin Fried, the 1st and 2nd place finishers, respectively, in the 2017 World Diplomacy Convention (widely considered the world championship for full-press Diplomacy) to play games against six copies of our agent. The purpose was to determine whether the human experts could discover exploitable weaknesses in the bot.

The humans played games against three types of bots: DipNet (Paquette et al., 2019) (with temperature set to 0.5), our blueprint agent (with temperature set to 0.5), and SearchBot. In total, the participants played 35 games against each bot; each of the seven powers was controlled by a human player five times, while the other six powers were controlled by identical copies of the bot. The performance of the humans is shown in Table 4. While the sample size is relatively small, the results suggest that our agent is less exploitable than prior bots. Our improvements to the supervised learning policy reduced the humans' score from 39.1% to 22.5%, which is itself a large improvement. However, the largest gains came from search. Adding search reduced the humans' score from 22.5% to 5.7%. A score below 14.3% means the humans are losing on average.

| Power | 1 Human vs. 6 DipNet | 1 Human vs. 6 Blueprint | 1 Human vs. 6 SearchBot |
|---|---|---|---|
| All Games | 39.1% | 22.5% | 5.7% |

Table 4: Average SoS score of one expert human playing against six bots under repeated play. A score less than 14.3% means the human is unable to exploit the bot. Five games were played for each power for each agent, for a total of 35 games per agent. For each power, the human first played all games against DipNet, then the blueprint model described in Section 3.1, and then finally SearchBot.

### 4.3.3 EXPLOITABILITY IN LOCAL SUBGAME

We also investigate the exploitability of our agent in the local subgame defined by a given board state, sampled actions, and assumed blueprint policy for the rest of the game. We simulate 7 games between a search agent and 6 DipNet agents, and plot the total exploitability of the average strategy of the search procedure as a function of the number of RM iterations, as well as the exploitability of the blueprint policies. Utilities $u_i$ are computed using Monte Carlo rollouts with the same (blueprint)

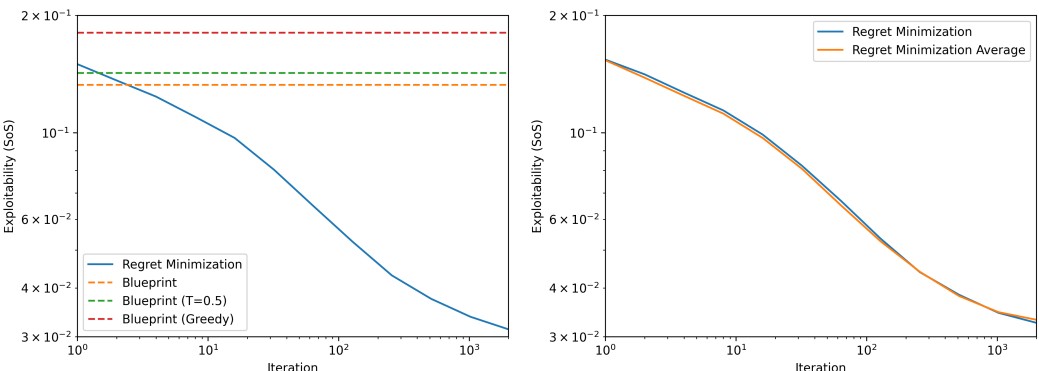

Figure 3: **Left:** Distance of the RM average strategy from equilibrium as a function of the RM iteration, computed as the sum of all agents' exploitability in the matrix game in which RM is employed. RM reduces exploitability, while the blueprint policy has only slightly lower exploitability than the uniform distribution over the 50 sampled actions used in RM (i.e. RM iteration 1). For comparison, our human evaluations used 256-2048 RM iterations, depending on the time per turn. **Right:** Comparison of convergence of individual strategies to the average of two independently computed strategies. The similarity of these curves suggests that independent RM computations lead to compatible equilibria. Note: In both figures, exploitability is averaged over all phases in 28 simulated games; per-phase results are provided in Appendix E.

rollout policy used during RM, and total exploitability for a joint policy $\pi$ is computed as $e(\pi) = \sum_i \max_{a_i \in \mathcal{A}_i} u_i(a_i, \pi_{-i}) - u_i(\pi)$. The exploitability curves aggregated over all phases are shown in Figure 3 (left) and broken down by phase in the Appendix.

In Figure 3 (right), we verify that the average of policies from multiple independent executions of RM also converges to an approximate Nash. For example, it is possible that if each agent independently running RM converged to a different incompatible equilibrium and played their part of it, then the joint policy of all the agents would not be an equilibrium. However, we observe that the exploitibility of the average of policies closely matches the exploitability of the individual policies.

## 5 CONCLUSIONS

No-press Diplomacy is a complex game involving both cooperation and competition that poses major theoretical and practical challenges for past AI techniques. Nevertheless, our AI agent achieves human-level performance in this game with a combination of supervised learning on human data and one-ply search using regret minimization. The massive improvement in performance from conducting search just one action deep matches a larger trend seen in other games, such as chess, Go, poker, and Hanabi, in which search dramatically improves performance. While regret minimization has been behind previous AI breakthroughs in purely competitive games such as poker, it was never previously shown to be successful in a complex game involving cooperation. The success of RM in no-press Diplomacy suggests that its use is not limited to purely adversarial games.

Our work points to several avenues for future research. SearchBot conducts search only for the current turn. In principle, this search could extend deeper into the game tree using counterfactual regret minimization (CFR) (Zinkevich et al., 2008). However, the size of the subgame grows exponentially with the depth of the subgame. Developing search techniques that scale more effectively with the depth of the game tree may lead to substantial improvements in performance. Another direction is combining our search technique with reinforcement learning. Combining search with reinforcement learning has led to tremendous success in perfect-information games (Silver et al., 2018) and more recently in two-player zero-sum imperfect-information games as well (Brown et al., 2020). Finally, it remains to be seen whether similar search techniques can be developed for variants of Diplomacy that allow for coordination between agents.

## ACKNOWLEDGEMENTS

We thank Kestas Kuliukas and the entire `webdiplomacy.net` team for their cooperation and for providing the dataset used in this research. We also thank Thomas Anthony, Spencer D., and Joshua M. for computing the agent's Ghost-Rating. We additionally thank Doug Moore and Marvin Fried for participating in the human exploiter experiments in Section 4.3.2, and thank Andrew Goff for providing expert commentary on the bot's play. Finally, we thank Jakob Foerster for helpful discussions.

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

# A  FULL IMITATION LEARNING MODEL ARCHITECTURE DESCRIPTION

In this section, we provide a full description of the model architecture used for the supervised Diplomacy agent.

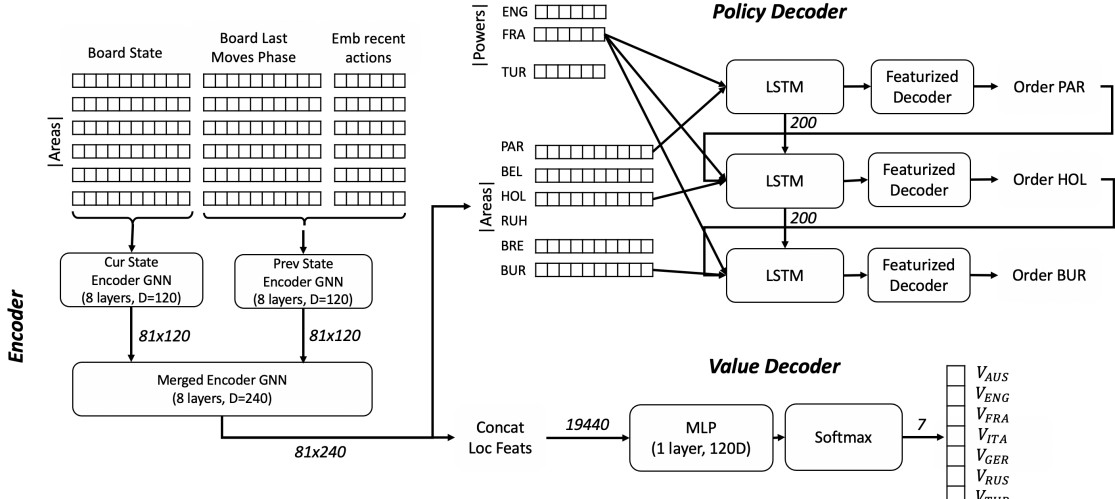

Figure 4: Architecture of the model used for imitation learning in no-press. Diplomacy.

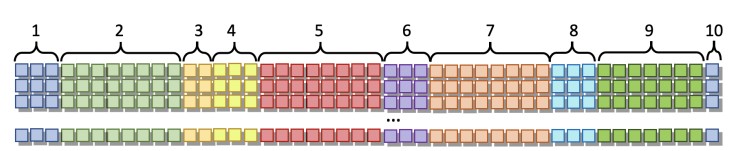

Figure 5: Features used for the board state encoding.

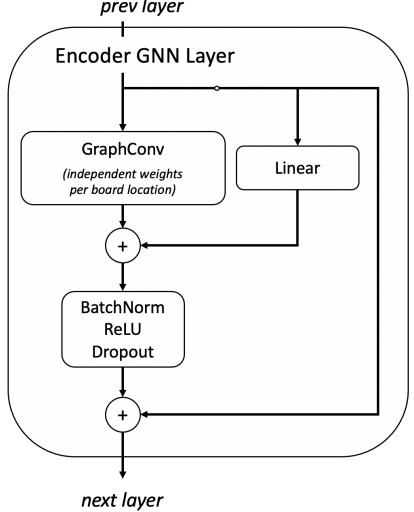

Figure 6: Architecture of an encoder GNN layer.

The model architecture is shown in Figure 4 . It consists of a shared encoder and separate *policy* and *value* decoder heads.

The input to the encoder is a set of features for each of the 81 board locations (75 regions + 6 coasts). The features provided for each location are shown in Figure 5. The first 7 feature types (35 features/location) match those of Paquette et al. (2019), and the last three feature types (in italics) are for global board features that are common across all locations: a 20-d embedding of the current season; the net number of builds for each power during a build phase; and a single feature identifying whether it is a no-press or full-press game.

The encoder has two parallel GNN trunks. The *Current State Encoder* takes the current board state as input, while the *Previous State Encoder* takes the board state features from the last movement phase, concatenated with a 20-dimensional feature at each location that is a sum of learned embeddings for each order that ocurred at this location during all phases starting at the last movement phase.

The current board state and previous board state are each fed to an 8-layer GNN, whose layer architecture is shown in Figure 6. The GraphConv operation consists of a separate learned linear transformation applied at each board location, followed by application of the normalized board adjacency matrix.[8] In parallel with the GraphConv is a linear layer that allows the learning of a feature hierarchy without graph smoothing. This is followed by batchnorm, a ReLU non-linearity, and dropout probability 0.4. The $81 \times 120$ output features from each of these GNNs are then concatenated and fed to another 8-layer GNN with dimensional 240. The output of this encoder are 240 features for each board location.

The value head takes as input all $81 \times 240$ encoder features concatenated into a single 19440-dimensional vector. An MLP with a single hidden layer followed by a softmax produces predicted sum-of-squares values for each power summing to 1. They are trained to predict the final SoS values with an MSE loss function.

The policy head auto-regressively predicts the policy for a particular power one unit at a time with a 2-layer LSTM of width 200. Unit orders are predicted in the global order used by Paquette et al. (2019). The input to the LSTM at each step is the concatenation of (a) the 240-d encoder features of the location whose orders are being predicted; (b) a 60-d embedding of the power whose policy is being computed; (c) an 80-d embedding of the predicted orders for the last unit.

A standard decoder matrix can be thought of as a learned vector for each valid order, whose dot product with the LSTM output predicts the logit of the order probability given the state. We featurize these order vectors as shown in Figure 7. The decoder vector for "PAR S RUH - BUR" , for example, would be the sum of a learned vector for this order and learned linear transformations of (a) one-hot features for the order source (RUH), destination (BUR), type (S); and (b) the encoder features of the source (RUH) and destination (BUR) locations.

Build orders are different than other order types because the set of source locations for build orders is not known in advance. So the decoder needs to predict an *unordered set* of build orders during a build phase, which is tricky to compute correctly with an autoregressive decoder and teacher forcing. Luckily, since a power can only build in its 3-4 home supply centers, there are only 170 sets of build orders across all powers, so we just consider each possible *set* of builds as a single order, rather than predicting them one by one. We do not do featurized order decoding for build orders.

## B  COMPUTING PLAYER RATINGS IN HUMAN DATA

To compute player ratings, we used a regularized logistic outcome model. Specifically, we optimize a vector of ratings **s** by gradient descent that minimizes the regularized negative log likelihood loss

$$L(\mathbf{s}|\mathcal{D}) = \sum_{(i,j)\in\mathcal{D}} -\log \sigma(s_i - s_j) + \lambda|\mathbf{s}|_2$$

over a dataset $\mathcal{D}$ consisting of all pairs of players $(i, j)$ where player $i$ achieved a "better" outcome than player $j$ in a game. We found this approach led to more plausible scores than Elo (Elo, 1978) or TrueSkill (Herbrich et al., 2007) ratings.

---

[8]As noted in Anthony et al. (2020), this differs from a standard graph convolutional layer in that separate weights are used for each board location.

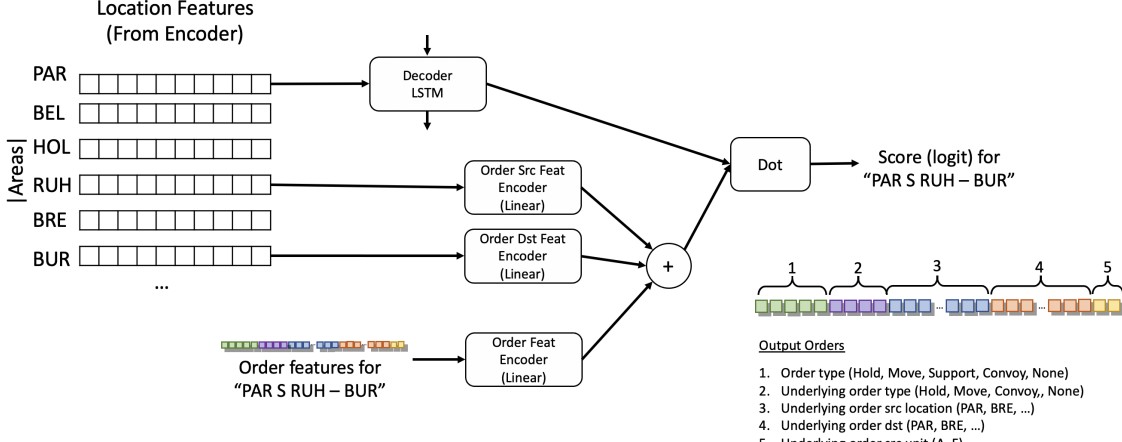

Figure 7: Illustration of featurized order decoding.

Paquette et al. (2019) took an orthogonal approach to filter poor players from the training data: they only trained on "winning" powers, i.e. those who ended the game with at least 7 SCs. This filtering is sensible for training a policy for play, but is problematic for training policies for search. In a general-sum game, it is crucial for the agent to be able to predict the empirical distribution of actions even for other agents who are destined to lose.

## C  SEARCH EXAMPLE

The listing below shows the policy generated by one run of our search algorithm for the opening move of Diplomacy when running for 512 iterations and considering 8 possible actions per power.

For each possible action, the listing below shows

probs : The probability of the action in the final strategy.

bp_p : The probability of the action in the blueprint strategy.

avg_u : The average predicted sum-of-squares utility of this action.

orders : The orders for this action

```
AUSTRIA avg_utility=0.15622
  probs    bp_p     avg_u    orders
  0.53648  0.13268  0.15697  ('A VIE - TRI', 'F TRI - ALB', 'A BUD - SER')
  0.46092  0.52008  0.14439  ('A VIE - GAL', 'F TRI - ALB', 'A BUD - SER')
  0.00122  0.03470  0.14861  ('A VIE - TRI', 'F TRI - ALB', 'A BUD - GAL')
  0.00077  0.03031  0.11967  ('A VIE - BUD', 'F TRI - ALB', 'A BUD - SER')
  0.00039  0.05173  0.11655  ('A VIE - GAL', 'F TRI S A VEN', 'A BUD - SER')
  0.00015  0.04237  0.12087  ('A VIE - GAL', 'F TRI H', 'A BUD - SER')
  0.00007  0.14803  0.09867  ('A VIE - GAL', 'F TRI - VEN', 'A BUD - SER')
  0.00000  0.04009  0.03997  ('A VIE H', 'F TRI H', 'A BUD H')
ENGLAND avg_utility=0.07112
  probs    bp_p     avg_u    orders
  0.41978  0.20069  0.07151  ('F EDI - NTH', 'F LON - ENG', 'A LVP - YOR')
  0.34925  0.29343  0.07161  ('F EDI - NTH', 'F LON - NTH', 'A LVP - YOR')
  0.10536  0.06897  0.07282  ('F EDI - NTH', 'F LON - ENG', 'A LVP - WAL')
  0.07133  0.36475  0.07381  ('F EDI - NWG', 'F LON - NTH', 'A LVP - EDI')
  0.05174  0.01649  0.07202  ('F EDI - NTH', 'F LON - ENG', 'A LVP - EDI')
  0.00249  0.00813  0.06560  ('F EDI - NWG', 'F LON - NTH', 'A LVP - WAL')
  0.00006  0.00820  0.06878  ('F EDI - NWG', 'F LON - ENG', 'A LVP - EDI')
  0.00000  0.03933  0.03118  ('F EDI H', 'F LON H', 'A LVP H')
FRANCE avg_utility=0.21569
  probs    bp_p     avg_u    orders
  0.92038  0.09075  0.21772  ('F BRE - MAO', 'A PAR - GAS', 'A MAR - BUR')
  0.06968  0.42617  0.18878  ('F BRE - MAO', 'A PAR - BUR', 'A MAR S A PAR - BUR')
  0.00917  0.07987  0.16941  ('F BRE - MAO', 'A PAR - PIC', 'A MAR - BUR')
  0.00049  0.05616  0.16729  ('F BRE - ENG', 'A PAR - BUR', 'A MAR - SPA')
  0.00023  0.17040  0.17665  ('F BRE - MAO', 'A PAR - BUR', 'A MAR - SPA')
  0.00004  0.04265  0.18629  ('F BRE - MAO', 'A PAR - PIC', 'A MAR - SPA')
  0.00001  0.09291  0.15828  ('F BRE - ENG', 'A PAR - BUR', 'A MAR S A PAR - BUR')
```

```
    0.00000   0.04109   0.06872   ('F BRE H', 'A PAR H', 'A MAR H')
GERMANY avg_utility=0.21252
  probs     bp_p      avg_u       orders
  0.39050   0.01382   0.21360   ('F KIE - DEN', 'A MUN - TYR', 'A BER - KIE')
  0.38959   0.02058   0.21381   ('F KIE - DEN', 'A MUN S A PAR - BUR', 'A BER - KIE')
  0.16608   0.01628   0.21739   ('F KIE - DEN', 'A MUN H', 'A BER - KIE')
  0.04168   0.21879   0.21350   ('F KIE - DEN', 'A MUN - BUR', 'A BER - KIE')
  0.01212   0.47409   0.21287   ('F KIE - DEN', 'A MUN - RUH', 'A BER - KIE')
  0.00003   0.05393   0.14238   ('F KIE - HOL', 'A MUN - BUR', 'A BER - KIE')
  0.00000   0.16896   0.13748   ('F KIE - HOL', 'A MUN - RUH', 'A BER - KIE')
  0.00000   0.03355   0.05917   ('F KIE H', 'A MUN H', 'A BER H')
ITALY avg_utility=0.13444
  probs     bp_p      avg_u       orders
  0.41740   0.19181   0.13609   ('F NAP - ION', 'A ROM - APU', 'A VEN S F TRI')
  0.25931   0.07652   0.12465   ('F NAP - ION', 'A ROM - VEN', 'A VEN - TRI')
  0.13084   0.29814   0.12831   ('F NAP - ION', 'A ROM - VEN', 'A VEN - TYR')
  0.09769   0.03761   0.13193   ('F NAP - ION', 'A ROM - APU', 'A VEN - TRI')
  0.09412   0.16622   0.13539   ('F NAP - ION', 'A ROM - APU', 'A VEN H')
  0.00034   0.05575   0.11554   ('F NAP - ION', 'A ROM - APU', 'A VEN - PIE')
  0.00028   0.13228   0.10953   ('F NAP - ION', 'A ROM - VEN', 'A VEN - PIE')
  0.00000   0.04167   0.05589   ('F NAP H', 'A ROM H', 'A VEN H')
RUSSIA avg_utility=0.06623
  probs     bp_p      avg_u       orders
  0.64872   0.05988   0.06804   ('F STP/SC - FIN', 'A MOS - UKR', 'A WAR - GAL', 'F SEV - BLA')
  0.28869   0.07200   0.06801   ('F STP/SC - BOT', 'A MOS - STP', 'A WAR - UKR', 'F SEV - BLA')
  0.04914   0.67998   0.05929   ('F STP/SC - BOT', 'A MOS - UKR', 'A WAR - GAL', 'F SEV - BLA')
  0.01133   0.01147   0.05023   ('F STP/SC - BOT', 'A MOS - SEV', 'A WAR - UKR', 'F SEV - RUM')
  0.00120   0.02509   0.05008   ('F STP/SC - BOT', 'A MOS - UKR', 'A WAR - GAL', 'F SEV - RUM')
  0.00064   0.09952   0.05883   ('F STP/SC - BOT', 'A MOS - STP', 'A WAR - GAL', 'F SEV - BLA')
  0.00027   0.01551   0.04404   ('F STP/SC - BOT', 'A MOS - SEV', 'A WAR - GAL', 'F SEV - RUM')
  0.00000   0.03655   0.02290   ('F STP/SC H', 'A MOS H', 'A WAR H', 'F SEV H')
TURKEY avg_utility=0.13543
  probs     bp_p      avg_u       orders
  0.82614   0.25313   0.13787   ('F ANK - BLA', 'A SMY - ARM', 'A CON - BUL')
  0.14130   0.00651   0.12942   ('F ANK - BLA', 'A SMY - ANK', 'A CON - BUL')
  0.03080   0.61732   0.12760   ('F ANK - BLA', 'A SMY - CON', 'A CON - BUL')
  0.00074   0.01740   0.11270   ('F ANK - CON', 'A SMY - ARM', 'A CON - BUL')
  0.00069   0.05901   0.12192   ('F ANK - CON', 'A SMY - ANK', 'A CON - BUL')
  0.00030   0.00750   0.11557   ('F ANK - CON', 'A SMY H', 'A CON - BUL')
  0.00001   0.00598   0.10179   ('F ANK S F SEV - BLA', 'A SMY - CON', 'A CON - BUL')
  0.00001   0.03314   0.04464   ('F ANK H', 'A SMY H', 'A CON H')
```

# D  RL DETAILS

Each action $a$ in the MDP is a sequence of orders $(o_1, \ldots, o_t)$. The probability of the order $a$ under policy $\pi_\theta$ is defined by an LSTM in auto regressive fashion, i.e., $\pi_\theta(a) = \prod_{i=1}^{t}(\pi_\theta(o_i|o_1 \ldots o_{i-1}))$. To make training more stable, we would like to prevent entropy $H(\pi_\theta) := -E_{a \sim \pi_\theta} \log(\pi_\theta(a)$ from collapsing to zero. The naive way to optimize the entropy of the joint distribution is to use a sum of entropies for each individual order, i.e., $\frac{d}{d\theta}H(\pi_\theta(\bullet)) \approx \sum_{i=1}^{t} \frac{d}{d\theta}H(\pi_\theta(\bullet|o_1 \ldots o_{i-1}))$. However, we found that this does not work well for our case probably because there are strong correlations between orders. Instead we use an unbiased estimate of the joint entropy that is agnostic to the size of the action space and requires only to be able to sample from a model and to adjust probabilities of the samples.

**Statement 1.** *Let $\pi_\theta(\bullet)$ be a probability distribution over a discrete set $A$, such that $\forall a \in A\ \pi_\theta(a)$ is a smooth function of a vector of parameters $\theta$. Then*

$$\frac{d}{d\theta}\left(H(\pi_\theta(\bullet))\right) = -E_{a \sim \pi_\theta}(1 + \log \pi_\theta(a))\frac{d}{d\theta}\log \pi_\theta(a).$$

*Proof.* Proof is similar to one for REINFORCE:

$$
\begin{aligned}
\frac{d}{d\theta}\left(H(\pi_\theta(\bullet))\right) &= \frac{d}{d\theta}\left(-E_{a\sim\pi_\theta}\log\pi_\theta(a)\right) \\
&= -\frac{d}{d\theta}\left(\sum_{a\in A}\pi_\theta(a)\log\pi_\theta(a)\right) \\
&= -\sum_{a\in A}\left(\frac{d}{d\theta}\pi_\theta(a)\log\pi_\theta(a)\right) \\
&= -\sum_{a\in A}\left(\pi_\theta(a)\frac{d}{d\theta}\log\pi_\theta(a) + \log\pi_\theta(a)\frac{d}{d\theta}\pi_\theta(a)\right) \\
&= -\sum_{a\in A}\left(\pi_\theta(a)\frac{d}{d\theta}\log\pi_\theta(a) + \log\pi_\theta(a)\pi_\theta(a)\frac{d}{d\theta}\log\pi_\theta(a)\right) \\
&= -\sum_{a\in A}\pi_\theta(a)\left((1+\log\pi_\theta(a))\frac{d}{d\theta}\log\pi_\theta(a)\right) \\
&= -E_{a\sim\pi_\theta}\left((1+\log\pi_\theta(a))\frac{d}{d\theta}\log\pi_\theta(a)\right).
\end{aligned}
$$

□

## E   SUBGAME EXPLOITABILITY RESULTS

Figure 8 plots the total exploitability of joint policies computed by RM in the subgame used for equilibrium search at each phase in 7 simulated Diplomacy games.

## F   PLAYING ACCORDING TO THE FINAL ITERATION'S POLICY IN RM

As described in Section 2.2, in RM it is the *average* policy over all iterations that converges to an equilibrium, not the final iteration's policy. Nevertheless, in our experiments we sample an action from the final iteration's policy. This technique has been used successfully in past poker agents (Brown & Sandholm, 2019b). At first glance this modification may appear to increase exploitability because the final iteration's policy is not an equilibrium and in practice is often quite pure. However, in this section we provide evidence that sampling from the final iteration's policy when using a Monte Carlo version of RM may *lower* exploitability compared to sampling from the average policy, and conjecture an explanation.

Critically, in the sampled form of RM the final iteration's policy depends on the random seed, which is unknown to the opponents (while we assume opponents have access to our code, we assume they do not have access to our run-time random seeds). Therefore, the opponent is unlikely to be able to predict the final iteration's policy. Instead, from the opponent's perspective, the policy we play is sampled from an average of final-iteration policies from all possible random seeds. Due to randomness in the behavior of sampled RM, we conjecture that this average has low exploitability.

To measure whether this is indeed the case, we ran sampled RM for 256 iterations on two Diplomacy situations (the first turn in the game and the second turn in the game) using 1,024 different random seeds, and averaged together the final iteration of each run. We also ran sampled RM on two random matrix games with 10,000 different random seeds. The results, shown in Table 5, confirm that averaging the final iteration over multiple random seeds leads to low exploitability.

We emphasize that while we measured exploitability in these experiments by running sampled regret matching over multiple random seeds, we only need to run it for a single random seed in order to achieve this low exploitability because the opponents do not know which random seed we use. Unfortunately, measuring the exploitability of the final iteration over 1,024 random seeds is about 1,024x as expensive as measuring the exploitability of the average strategy over all iterations (and

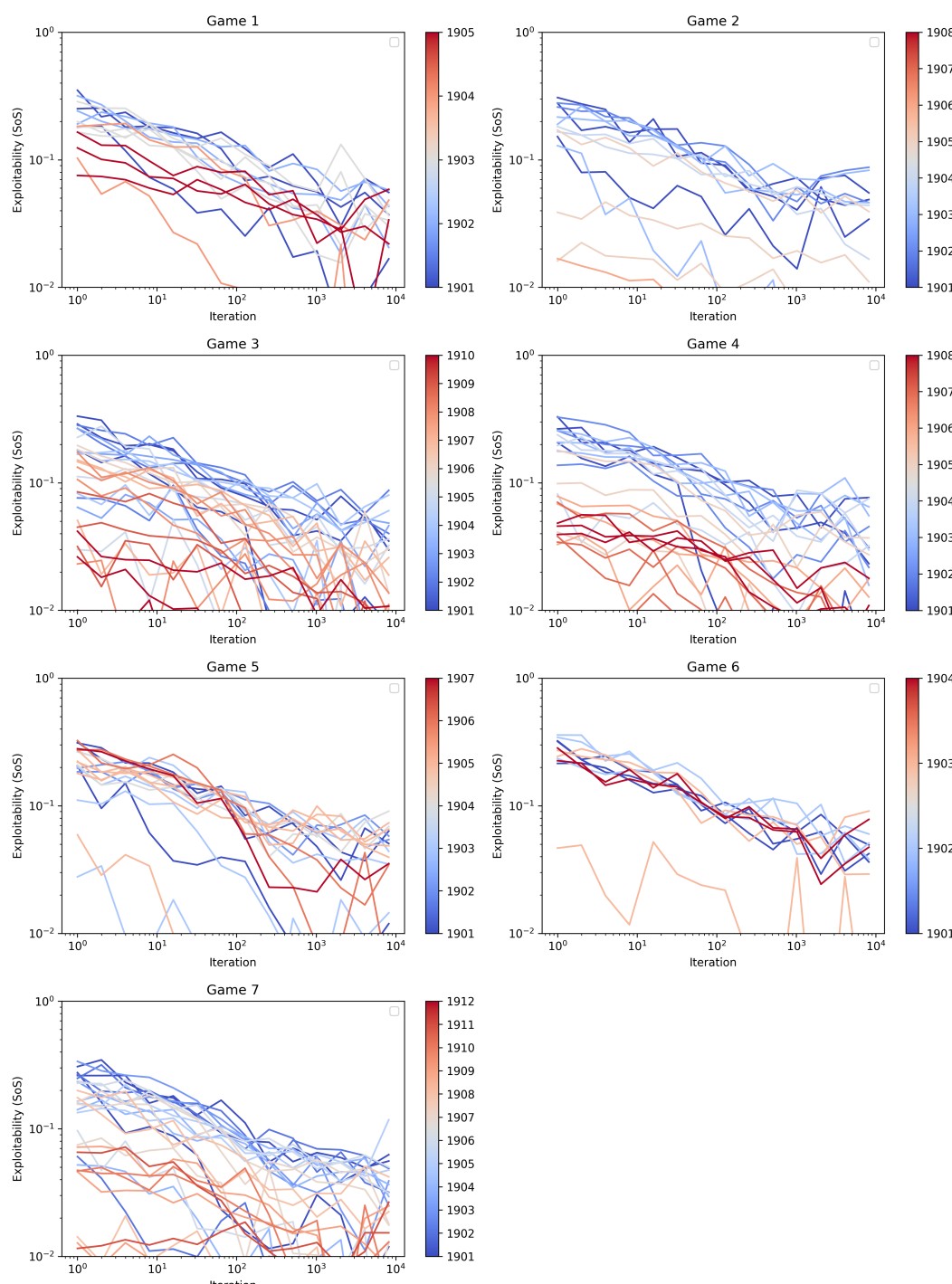

Figure 8: Exploitability as a function of RM iteration at each phase of 7 simulated games with our search agent (until the game ends or the search agent is eliminated). Aggregate results in Figure 3.

even 1,024 seeds would provide only an approximate upper bound on exploitability, since computing the true exploitability would require averaging over all possible random seeds). Therefore, in order to keep the computation tractable, we measure the exploitability of the average strategy of single runs in Figure 3.

| Method | Subgame 1 | Subgame 2 | 10x10 Random | 100x100 Random |
|---|---|---|---|---|
| Ave. of Final Policies | 0.00038 | 0.0077 | 0.019 | 0.063 |
| Ave. of Ave. Policies | 0.00027 | 0.0074 | 0.035 | 0.092 |
| Single Ave. Policy | 0.00065 | 0.0140 | 0.078 | 0.225 |
| Single Final Policy | 0.00880 | 0.0580 | 0.478 | 0.706 |

Table 5: Exploitability of 256 iterations of RM using either the final iteration's policy or the average iteration's policy, and either using the policy from a single run or the policy average over multiple runs. Subgame 1 and Subgame 2 are subgames from Diplomacy S1901 and F1901, respectively. 10x10 Random and 100x100 Random are random matrix two-player zero-sum games with entries having value in $[0, 1)$ sampled uniformly randomly. For Subgame 1 and Subgame 2, "Ave. of Final Policies" and "Ave. of Ave. Policies" are the average of 1,024 runs with different random seeds. For 10x10 Random and 100x100 Random, 10,000 seeds were used.

## G DETAILS ON EXPERIMENTAL SETUP

Most games on `webdiplomacy.net` were played with 24-hour turns, though the agent also played in some "live" games with 5-minute turns. Different hyperparameters were used for live games versus non-live games. In non-live games, we typically ran RM for 2,048 iterations with a rollout length of 3 movement phases, and set $M$ (the constant which is multiplied by the number of units to determine the number of subgame actions) equal to 5. This typically required about 20 minutes to compute. In live games (including games in which one human played against six bots) and in games against non-human opponents, we ran RM for 256 iterations with a rollout length of 2 movement phases, and set $M$ equal to 3.5. This typically required about 2 minutes to compute. In all cases, the temperature for the blueprint in rollouts was set to 0.75.

| Power | Score | Human Mean | Games | Wins | Draws | Losses |
|---|---|---|---|---|---|---|
| All Games | 26.6% $\pm$ 3.2% | 14.3% | 116 | 16 | 43 | 57 |
| Normalized By Power | 26.9% $\pm$ 3.3% | 14.3% | 116 | 16 | 43 | 57 |
| Austria | 31.4% $\pm$ 9.4% | 11.0% | 17 | 3 | 6 | 8 |
| England | 38.0% $\pm$ 10.1% | 12.6% | 16 | 4 | 6 | 6 |
| France | 19.0% $\pm$ 6.2% | 16.9% | 19 | 1 | 8 | 10 |
| Germany | 36.8% $\pm$ 8.0% | 15.0% | 16 | 2 | 10 | 4 |
| Italy | 31.5% $\pm$ 10.5% | 11.6% | 14 | 3 | 5 | 6 |
| Russia | 17.6% $\pm$ 7.6% | 14.8% | 18 | 2 | 4 | 12 |
| Turkey | 14.6% $\pm$ 7.0% | 18.2% | 16 | 1 | 4 | 11 |

Table 6: Average SoS score of our agent in anonymous games against humans on `webdiplomacy.net`. Average human performance is 14.3%. Score in the case of draws was determined by the rules of the joined game. The $\pm$ shows one standard error. Average human performance was calculated based on SoS scoring of historical games on `webdiplomacy.net`.

The experiments on `webdiplomacy.net` occurred over a three-month timespan, with games commonly taking one to two months to complete (players are typically given 24 hours to act). Freezing research and development over such a period would have been impractical, so our agent was not fixed for the entire time period. Instead, serious bugs were fixed, improvements to the algorithm were made, and the model was updated.

In games on `webdiplomacy.net`, a draw was submitted if SearchBot did not gain a center in two years, or if the agent's projected sum-of-squares score was less than the score it would achieve by an immediate draw. Since there was no way to submit draws through the `webdiplomacy.net` API, draws were submitted manually once the above criteria was satisfied. Games against other bots, shown in Table 3, automatically ended in a draw if no player won by 1935.

| Power | 1 Human vs. 6 DipNet | 1 Human vs. 6 Blueprint | 1 Human vs. 6 SearchBot |
|---|---|---|---|
| All Games | 39.1% | 22.5% | 5.7% |
| Austria | 40% | 20% | 0% |
| England | 28.4% | 20% | 0% |
| France | 20% | 4.3% | 40% |
| Germany | 40% | 33% | 0% |
| Italy | 60% | 20% | 0% |
| Russia | 40% | 20% | 0% |
| Turkey | 45.1% | 40% | 0% |

Table 7: Average SoS score of one expert human playing against six bots under repeated play. A score less than 14.3% suggests the human is unable to exploit the bot. Five games were played for each power for each agent, for a total of 35 games per agent. For each power, the human first played all games against DipNet, then the blueprint model described in Section 3.1, and then finally SearchBot.

## H  QUALITATIVE ASSESSMENT OF SEARCHBOT

Qualitatively, we observe that SearchBot performs particularly well in the early and mid game. However, we observe that it sometimes struggles with endgame situations. In particular, when it is clear that one power will win unless the others work together to stop it, SearchBot will sometimes continue to attack its would-be allies. There may be multiple contributing factors to this. One important limitation, which we have verified in some situations, is that the sampled subgame actions may not contain any action that could prevent a loss. This is exacerbated by the fact that players typically control far more units in the endgame and the number of possible actions grows exponentially with the number of units, so the sampled subgame actions contain a smaller fraction of all possible actions. Another possible contributing factor is that the state space near the end of the game is far larger, so there is relatively less data for supervised learning in this part of the game. Finally, another possibility is that because the network only has a dependence on the state of the board and the most recent player actions, it is unable to sufficiently model the future behavior of other players in response to being attacked.

Although the sample size is small, the results suggest that SearchBot performed particularly well with the central powers of Austria, Germany, and Italy. These powers are considered to be the most difficult to play by humans because they are believed to require an awareness of the interactions between all players in the game.

## I  ANECDOTAL ANALYSIS OF GAMES BY EXPERT HUMAN

Below we present commentary from Andrew Goff, three-time winner of the world championship for Diplomacy. The comments are lightly edited for clarity. This is intended purely as anecdotal commentary from a selection of interesting games.

### GAME 1

http://webdiplomacy.net/board.php?gameID=319258 (bot as Germany)

The bot is doing a lot of things right here, particularly if they are playing for an 18. The opening to Tyrolia is stronger than most players realise, and the return to defend Munich is exactly how to play it. Building 2 fleets is hyper-aggressive but in Gunboat it is a good indication move, particularly after the early fight between England and France. 1902 the board breaks very favourably for Germany, but its play is textbook effective. The bounce in Belgium in Fall is a very computer move... 9/10 times it bounces as it does in the game, but the 1/10 times it is a train wreck, as England retreats to Holland and France is angry so it is a risk human players don't often take.

1903 is a bit wilder. The play in the North is solid (Scandinavia is more important than England centres – I agree) but the move to Burgundy seems premature. In gunboat it is probably fine, but why not Ruhr then Belgium? Why not give France greater opportunity to suicidally help you against

England and Italy. I suspect this is an artefact of being trained against other AI that would not do that?

1904 has a nice convoy (to Yorkshire) and strong moves over the line into Russia. It is all good play, but in fairness the defences put up by the opponents are weak. After that the next few years are good consolidations – a little fidgety in Norway but not unreasonable. 1907 see Turkey dot[9] Russia and from here you should expect an 18 – a terrible move by Turkey. It should be all over now and it is just clean-up. Even as Turkey and Russia patch things up quite quickly, Turkey picks off an Italian centre and now it really is all over. France collapses, Germany cleans up professionally, and score one for AI.

Overall, Germany's play was aggressive early and then clinical late. Some pretty bad errors by Turkey gave Germany the chance, and they took it. 9/10

## GAME 2

http://webdiplomacy.net/board.php?gameID=319189 (bot as Italy)

Spoiler Alert: this one isn't so good. The opening is sleepy and the attack on Vienna in 1902 is just wrong against human players. Then the whole rest of the board is playing terribly too – Austria goes for a Hail Mary by attacking their only potential ally. . . Turkey tries a convoy that if it had worked would suggest collusion. . . this is all terrible. Now everyone is fighting everyone and meanwhile Germany and France have cleaned up England. By 1905 at least the AI has given up attacking Austria for the time being – picking the competent over the nonsense – and the day is saved as France and Germany descend into brawling over the spoils.

It doesn't last. Another year another flip/flop. The same attempted trick again (move to Pie then up to Tyrol). Italy requires patience and the AI is showing none at all. Austria gets around to Armenia – they're playing OK tactically but they have a mad Italian hamstringing them both. Meanwhile France is now officially on top and will present a solo threat soon. The start of 1908 sees Italy finally making some progress against Austria, but France is too big now. . . it is all strategically a mess from 1902 onward and it shows. As a positive though, some of the tactics here are first rate. . . Nap – Alb is tidy, the progress that is being made against Austria is good in isolation.

Then follows 8 years of stalemate tactics none of which is interesting. Italy played well, was lucky Turkey didn't take their centres at the death, and that's all there is.

Overall, I'd say this looked like a human player. And just like most human players: rubbish at Italy. 3/10

## GAME 3

http://webdiplomacy.net/board.php?gameID=318752 (bot as Austria)

I was about to say 1901 was all boring and then the AI in Austria has built a fleet. That's wild. With Italy attacking France it isn't unreasonable. . . but still. Anyhow as the game continues into 1903 he deploys that fleet beautifully. Italy just doesn't care about it – which is strategically very poor by them. The weird thing here is that Austria seems more intent on attacking Russia than Turkey. If you have the second fleet then surely Turkey is the objective, otherwise you're splitting forces a lot. In gunboat it is. . . passable. In non-Gunboat this is all-but suicidal. If RT get their heads together they'll belt Austria, but as we see in this game they don't so Austria does well from it. It's similar to game 1 in that this is the most aggressive way to play.

One thing really stands out here and that is that the AI is making the moves that are most likely to result in an error by the opponent (See: Turkey, S1904). This is present elsewhere but this turn captures it beautifully. Why did Smyrna go to Con? What a shocker. But without moves that can take advantage of errors like this it doesn't matter. Italy also makes a key mistake here: Mar holds instead of → Gas. They've lost momentum on the turn Austria gains a lot. In Fall Austria builds 3 armies and Italy gets nothing and the chance is there to solo.

---

[9]To "dot" a player in Diplomacy means to attack an ally for a gain of only one center.

The stab comes immediately and appropriately, and Austria has a fleet in the Black Sea and Aegean. Italy is a turn too late into the Northern French centres. There is a subtle problem to the AI's play here though, not seen in the Germany game. There's no effort to move over the stalemate positions. At this point in the German game it moved aggressively into Russia. . . here it ignores Germany and aims for the Russian centres again and that shows a lack of awareness of where the 18th centre must come from.

The game continues and Austria cleans up the Eastern centres very effectively. I like ignoring the German move to Tyrolia and Vienna – not very human but also very correct (as we will see – there's nothing there to support it so it just gets yeeted in Spring 1908. However, Spring 1908 has a tactical error (one of the few outright wrong moves I've seen!) where:

Gre s Adr – Ion; Aeg s Adr – Ion; Adr – Ion.

You only need one support and Adriatic is more useful than Greece. . . so better is: Gre – Ion; Aeg s Gre – Ion; Adr – Apu.

This allows Nap – Rom; Apu – Nap next turn. A very odd mistake as this is the kind of detail I'd expect AI to do better than human. Ultimately it doesn't matter so much as Italy is going to concede Tunis to set up a rock-solid defence of Spa/Mar. But still.

And now. . . . The ultimate "Not good". For four turns in a row England fails to defend StP correctly and the AI misses it and therefore the chance to get 18. England is attacking Livonia with 3 and moving only one unit into StP behind it. Germany is not cutting Prussia. So:

Pru s Lvn, War s Lvn, Lvn s Mos – StP, Mos – StP

And Austria gets an 18. This is a really bad miss by the AI (not to mention England). As in every decent human player or better would have got an 18 here.

Overall, Austria played a really good game but didn't do some of the things I had high hopes for after game 1 – they didn't move over the stalemate and they didn't play as tactically precisely. Then the miss at the end. . . makes me feel good about being human. 7/10 until the last move, but in the end 5/10.

GAME 4

http://webdiplomacy.net/board.php?gameID=318726 (bot as Russia)

StP – Fin. Cute. The whole board just completely opens in the other direction from Russia so they get 7 centres at the end of 1901. I wish that happened to me when I played Russia. Of course, now everyone moves against Russia so we'll see how the AI defends. Italy has just demolished France, which makes this a strange game strategically too.

F1903 you could write a whole article on. What it looks like is Russia and Turkey successfully signalling "alliance" and that is impressive for an AI. The tactics are good – ignoring Warsaw to get Budapest, but the bounce in Sweden and drop to Baltic not so much. Why England didn't support Den – Swe is a mystery and it is probably all over for Russia if they do. But they didn't so play on.

The English play from here on is just great. F1905 the move through Swe – Den is the height of excellence. It obviously helped Russia too, but from an awful position there is now real potential. . . what a great move. Focussing on the AI – it just gets the tactics right. It pushes Germany back and nails the timing to take Vienna. But then it builds a Fleet in Sev. No no no. An Army in Sev is better tactically and it also doesn't tip off the Turk. Just as it got the signalling right earlier it got it exactly wrong here.

In the end the Turk starts blundering under pressure so as the kids say: "Whatever". The F1906 move of Adr – Alb instead of supporting Tri is egregiously bad and turns a bad position into a terrible one. They proceed to self-immolate and Russia and Italy are cruising now, and England's play remains first class. The clean-up of Turkish dots is effective, and then the rest is quite boring.

The biggest thing that struck me about this game (apart from England's brilliant recovery) was that Russia didn't manage unit balance well. Disbanding their only Northern fleet was a long-term error even if it made the most sense at the time; build F Sev instead of A Sev. . . and then ironically being left with too few Southern Fleets anyhow. Russia is hard like that, but the AI played no better than

a good human player on that score. To give a practical example – England never had enough Fleets to blow up F Bal, so if you swap A Pru for F Bal then this is a real chance at an 18.

Overall, an interesting game characterised by the best and worst of human play (England/Turkey) and a strong but flawed game by the AI. 7/10.

GAME 5

`http://webdiplomacy.net/board.php?gameID=318335` (bot as France)

Right off the bat I am biased as I love this kind of opening as France. The Fleet in English Channel is not good news though. We get to see defence from the AI, and frankly the next 5 years are boring as we watch a clinical tactics lesson. About F1906/S1907 there needed to be a pivot to help Italy and stop attacking Germany in order to stop the 18, and it just never came. What had been a good display of defence turned into a disaster of AI greed and lack of appreciation for the risk of another player getting an 18. Not only did the AI allow the 18 to happen, but it can fairly reasonably be blamed for it. I think this is something in the reward conditions you might need to look at, as this play makes no sense – it isn't even worth digging into it further here – it is just flat out strategically wrong.

Overall, good tactics but terrible game awareness and strategy. 2/10.

GAME 6

`http://webdiplomacy.net/board.php?gameID=318329` (bot as England)

This is more like it. In fact, I think this is the best game of Gunboat I've seen in ages. There's a pressure and inevitability about it from about 1903 onward... Turkey is aggravating everyone near them, and England is moving in second and having people just let it take their centres – it outplays Germany (and punishes F Kie – Hol by bouncing Denmark... heartily approved), then Russia and Austria just walk out of their centres for it. This makes the attack against France with Italy easy. The game should draw at this point, but Turkey just can't help themselves and stabs Italy when England is already on 15 – terrible, inexcusable play. Italy doesn't even turn around to fight Turkey, but it doesn't matter as there are just too many disbands and Turkey's armies are not fast enough to get to the front.

It's an 18 and it is going on 34. Solo victories are usually caused by someone else making terrible play and this is no exception – Turkey was awful. But they also require precise play by the winner. The AI delivered. Even though this is not the style I prefer to play England as, every move makes sense. The builds are precise, the critical moments are executed perfectly (For example: S1908 convoy Lon – Pic, then ignoring Holland to force English Channel... that's a high level play right there) and the sense of not attacking Russia when given the chance a few times is [chef's kiss]. Then, above all, the presence to stop building fleets and get the armies rolling – this is the inflection point of great England play and it caught Turkey off guard – as they built another fleet, then an army that moved to Armenia, England rushed the middle of the board and Turkey just can't cover all the gaps alone. This opened the door to "The Mistake" and Turkey walked right through it.

This is the easiest game to analyse because there's just no glaring mistakes by the AI and it takes correct advantage of the situation throughout. This is the only game that made me think the AI could eventually consistently beat human players. As impressive as the German game was it was hyper-win-oriented so it is high risk/high reward. This... this is just excellent Gunboat Diplomacy from start to finish.

Overall – superb. 10/10.

GAME 7

`http://webdiplomacy.net/board.php?gameID=319187` (bot as Austria)

Sometimes you just get killed.

I saw the first four turns and thought "this will be over by 1905" but to my delight the AI fought it out bravely and also did its very best to try and get back in the game – rather than accepting being

tucked in Greece it made a run for a home centre and then did everything it could to get a second build. All to no avail in the end, but the play is hard to fault.

Obviously this is a shorter review as there really isn't anything to see here. But as a finishing thought it will be this challenge that an AI faces most when it starts playing with natural language as well as just gunboat. In gunboat it is just a crap shoot who gets attacked early and how vicious it is; in non-gunboat this is a factor very much in control of the player. I think the AI has proven tactically savvy enough to cut it once it gets a start, but will it get starts in non-gunboats?

Overall – just forget this game ever happened, the same as any good human player would. 5/10.

