# OpenReview forum: "Human-Level Performance in No-Press Diplomacy via Equilibrium Search"
_ICLR.cc/2021/Conference — ICLR 2021 Oral_

### Official Review · AnonReviewer4 · 2020-10-22
**Official Blind Review #4**

**Rating:** 8
**Confidence:** 4

**Review:**

This paper proposes a combination of imitation learning and search applied to the multiplayer, simultaneous-move game of no-press Diplomacy. While both techniques have been used before, even in concert, there are some domain-specific challenges: more than two players, simultaneous moves, and a very large branching factor per player. The authors use imitation of human play for value estimates and selection of candidate actions, and External Regret Matching to generate a one-step policy. Experimental validation showed the agent to have strong human-level performance, and was hard to exploit using other machine-learning techniques.

To the best of my knowledge, I agree with the author's claim that the imitation/search agent from this paper is the first to demonstrate human-level performance in the game of Diplomacy. This is a non-trivial result: Diplomacy is a large problem, and is thought to require fairly complicated interaction between players throughout a game, as well as the ability to handle the uncertainty of other player's moves. There is no claim of guaranteed performance across a general class of problems, but the paper's method seems general enough to apply elsewhere.

The writing was generally clear, although asking "could I duplicate these results?" did raise a few questions about some details.
The experimental results against humans seem well constructed, and do a reasonable job to support the claim that the agent demonstrates human-level play. The assorted exploitability-inspired experiments help support the notion that the agent's play is strong, rather than just doing well based on some peculiarity of human play.

---------- specific issues

section 2.2 "(which can also be represented as v_i(a_i, a_{-i})."
Closing parenthesis missing?

"ERM guarantees that R^t_i(a_i) in \mathcal{O}(\sqrt{t}). If R^t_i(a_i) grows sublinearly for all players' actions, as in ERM, then the average policy over all iterations converges to a NE in two-player zero-sum games and in general the empirical distribution of players' joint policies converges to a CCE as t approaches infinity."
How does the sampling interact with this? The regret bound is w.r.t. the probabilistic policy pi^t_i, but sampled a_{-i}. Connecting this bound to describe the joint behaviour (i.e., Nash eq'm) prThis paper proposes a combination of imitation learning and search applied to the multiplayer, simultaneous-move game of no-press Diplomacy. While both techniques have been used before, even in concert, there are some domain-specific challenges: more than two players, simultaneous moves, and a very large branching factor per player. The authors use imitation of human play for value estimates and selection of candidate actions, and External Regret Matching to generate a one-step policy. Experimental validation showed the agent to have strong human-level performance, and was hard to exploit using other machine-learning techniques.

To the best of my knowledge, I agree with the author's claim that the imitation/search agent from this paper is the first to demonstrate human-level performance in the game of Diplomacy. This is a non-trivial result: Diplomacy is a large problem, and is thought to require fairly complicated interaction between players throughout a game, as well as the ability to handle the uncertainty of other player's moves. There is no claim of guaranteed performance across a general class of problems, but the paper's method seems general enough to apply elsewhere.

The writing was generally clear, although asking "could I duplicate these results?" did raise a few questions about some details.
The experimental results against humans seem well constructed, and do a reasonable job to support the claim that the agent demonstrates human-level play. The assorted exploitability-inspired experiments help support the notion that the agent's play is strong, rather than just doing well based on some peculiarity of human play.

---------- specific issues

section 2.2 "(which can also be represented as v_i(a_i, a_{-i})."
Closing parenthesis missing?

"ERM guarantees that R^t_i(a_i) in \mathcal{O}(\sqrt{t}). If R^t_i(a_i) grows sublinearly for all players' actions, as in ERM, then the average policy over all iterations converges to a NE in two-player zero-sum games and in general the empirical distribution of players' joint policies converges to a CCE as t approaches infinity."
How does the sampling interact with this? The regret bound is w.r.t. the probabilistic policy pi^t_i, but sampled a_{-i}. Connecting this bound to describe the joint behaviour (i.e., Nash eq'm) proceeds easily when considering the joint pi^t. However, in this case the joint is only connected by the specific sampled actions. Does the statement about convergence to NE apply to sampled ERM (With high probability? In expectation?) or only to ERM?

Table 1
What are T=0.5 and T=0.1?  I assume they're the temperature parameter, for action selection in DipNet, but this isn't described until later.

Figure 1, "Sampling a single action leads to poor performance"
How is the graph intended to be interpreted? In the graph, 1 search action looks to be significantly better than the blueprint.

"M_i is a hyperparameter that is proportional to the number of units controlled by player i."
Proportional how? If power i controls n units, the algorithm will sample M_i actions? M_i*n actions? k*n actions for some hyperparameter k? Please clarify this sentence.
Are the samples select independently across units somehow, or just the top orders for the round (for that possibly very large number of actions)?

"We compute a policy for each agent by running the sampled algorithm described in Equation 1 and Equation 2".
Section 2.2 describes a few modifications, one of which would seem to modify equation 2. Maybe saying instead that it is running the sampled algorithm described in Section 2.2?
How many iterations are used?

"If our agent is an ESS (or Nash equilibrium), then a population of players all playing the agent's policy could not be 'invaded' by a different policy."
Wouldn't this only apply to ESS, not Nash eq'm in general?

Section 4.2.1
What does a sum-of-squares score of 0.54 or 0.42 represent? Are those scores good? Huge? Tiny? The drop is statistically significant. Is that drop of ~0.12 large? Small? Are we supposed to care about the drop? Or just how low they both are? Or just how low the SearchBot-clone result is?
It seems like this section would be helped by having a short summary statement of the conclusions the authors would like to draw.

Section 4.2.2
It was nice to see that the humans beat the human-imitation blueprint, while they struggled quite a bit against SearchBot. It might be worth specifically noting that not only was the blueprint imitation an improvement on DipNet in this setting, it was specifically the search method making the largest improvement.

Section 4.2.3
CFR->ERM?  The text talks about plotting versus the number of ERM iterations, while the figure talks about CFR.

Figure 3
What exactly is the average being talked about in the first paragraph? The averaging mention in Section 2.2 (which is not used during search in normal play)? The self-play strategy profile which is a combination of one strategy from each of the 7 runs (not really an average)?  Something else?
Is "average" in the first paragraph different than "average" in the right figure? The average line on the left doesn't seem to match the average line on the right: on the right, there seems to be a slight bump upwards at the end, which doesn't seem to be present on the left.

On the right plot, why not include all (or at least multiple) independent plots, to give a more complete picture of the variance?

Include some mention or mark indicating how many iterations are used during play? As is, I don't know how this graph relates to the search as used in SearchBot.oceeds easily when considering the joint pi^t. However, in this case the joint is only connected by the specific sampled actions. Does the statement about convergence to NE apply to sampled ERM (With high probability? In expectation?) or only to ERM?

Table 1
What are T=0.5 and T=0.1?  I assume they're the temperature parameter, for action selection in DipNet, but this isn't described until later.

Figure 1, "Sampling a single action leads to poor performance"
How is the graph intended to be interpreted? In the graph, 1 search action looks to be significantly better than the blueprint.

"M_i is a hyperparameter that is proportional to the number of units controlled by player i."
Proportional how? If power i controls n units, the algorithm will sample M_i actions? M_i*n actions? k*n actions for some hyperparameter k? Please clarify this sentence.
Are the samples select independently across units somehow, or just the top orders for the round (for that possibly very large number of actions)?

"We compute a policy for each agent by running the sampled algorithm described in Equation 1 and Equation 2".
Section 2.2 describes a few modifications, one of which would seem to modify equation 2. Maybe saying instead that it is running the sampled algorithm described in Section 2.2?
How many iterations are used?

"If our agent is an ESS (or Nash equilibrium), then a population of players all playing the agent's policy could not be 'invaded' by a different policy."
Wouldn't this only apply to ESS, not Nash eq'm in general?

Section 4.2.1
What does a sum-of-squares score of 0.54 or 0.42 represent? Are those scores good? Huge? Tiny? The drop is statistically significant. Is that drop of ~0.12 large? Small? Are we supposed to care about the drop? Or just how low they both are? Or just how low the SearchBot-clone result is?
It seems like this section would be helped by having a short summary statement of the conclusions the authors would like to draw.

Section 4.2.2
It was nice to see that the humans beat the human-imitation blueprint, while they struggled quite a bit against SearchBot. It might be worth specifically noting that not only was the blueprint imitation an improvement on DipNet in this setting, it was specifically the search method making the largest improvement.

Section 4.2.3
CFR->ERM?  The text talks about plotting versus the number of ERM iterations, while the figure talks about CFR.

Figure 3
What exactly is the average being talked about in the first paragraph? The averaging mention in Section 2.2 (which is not used during search in normal play)? The self-play strategy profile which is a combination of one strategy from each of the 7 runs (not really an average)?  Something else?
Is "average" in the first paragraph different than "average" in the right figure? The average line on the left doesn't seem to match the average line on the right: on the right, there seems to be a slight bump upwards at the end, which doesn't seem to be present on the left.

On the right plot, why not include all (or at least multiple) independent plots, to give a more complete picture of the variance?

Include some mention or mark indicating how many iterations are used during play? As is, I don't know how this graph relates to the search as used in SearchBot.

=-=  Comments after author discussion
(Minor) concerns have been addressed.  Thank you for the revisions.

---

> ### Author Response · Authors · 2020-11-20
> **Response**
>
> Thank you!
>
> We intend to open source the code.
>
> **Convergence of sampled ERM:**
> Yes, ERM with external sampling (i.e. sampling pi^-i) converges at O(sqrt(t)) with high probability. This is just a special case of external sampling MCCFR, whose convergence is derived in (Lanctot et al. NeurIPS-09). We have added a clarification in the text.
>
> We've changed Figure 1 to specify that 0.5 and 0.1 are temperatures.
>
> **"Sampling a single action leads to poor performance" inconsistent with graph:**
> Thanks for pointing this out. Performing “search” with a single action is a degenerate case, because no search can be performed. In practice, the search agent would sometimes sample all-holds and lose (SoS=0.08), so for #actions=1 in the figure we removed the all-holds action from the plausible set, leading to SoS=0.25. In retrospect, both of these numbers are potentially misleading to the reader so we updated the plot to start at #actions=2.
>
> **Regarding M_i:**
> We changed this description to specify that the number of subgame actions is M * k_i, where M is a constant (usually 5) and k_i is the number of units that power i controls. An action is a set of orders for all units.
>
> **“Maybe saying instead that it is running the sampled algorithm described in Section 2.2? How many iterations are used?”:**
> Thanks for the suggestion, we made this wording change. The number of iterations was between 256 and 4,096 depending on the type of game. Live games used 256 iters, while non-live games used up to 4,096.
>
> **“Wouldn't this only apply to ESS, not Nash eq'm in general?”:**
> That’s correct, thanks for pointing that out. Based on comments from Reviewer3, we removed the paragraph discussing ESS. However, if the reviewers feel it would be better to include the ESS discussion, we will include your correction.
>
> We have added your proposed changes for sections 4.2.1, 4.2.2, and 4.2.3.
>
> **“Figure 3 What exactly is the average being talked about in the first paragraph?”:**
> We are talking about the average policy over all iteration, which is what converges in theory to a NE in 2p0s games. However, as you point out, in practice we use the final iterate strategy. The final iterate strategy is not itself expected to be close to an equilibrium. However, the final iterate strategy depends on the random seed we use, which is unknown to the opponents (while we assume opponents have access to our code, we assume they don't know our run-time random seeds). Therefore, using the final iterate strategy is unlikely to be exploitable because the final iterate strategy is unpredictable. To confirm that this is indeed the case, we ran our sampled regret matching algorithm for 256 iterations on two Diplomacy situations (the first turn in the game and the second turn in the game) using 1,024 different random seeds, and averaged together the final iteration of each run. While the NashConv of the strategy on the final iteration of a single run was 8.8 * 10^-3 for the first turn and 5.8 * 10^-2 for the second turn, the NashConv when averaging over 1,024 different random seeds was 3.8 * 10^-4 for the first turn and 7.7 * 10^-3 for the second turn. For comparison, using the average strategy over all iterations for a single random seed had a NashConv of 6.5 * 10^-4 for the first turn and 1.4 * 10^-2 for the second turn. This shows using the final iteration’s strategy can result in low exploitability. (We also found that averaging the average strategy over 1,024 random seeds achieves a NashConv of 2.7 * 10^-4 on the first turn and 7.4 * 10^-3 on the second turn, which suggests that the NashConv numbers obtained from a single run are slightly higher than the true NashConv numbers.) We also verified that the same result holds in random 10x10 and 100x100 matrix games. We emphasize that while we *measured* NashConv in this experiment by running sampled regret matching for 1,024 different random seeds, we only need to run it for a *single* random seed in order to achieve this low exploitability because the opponents do not know which random seed we use. Unfortunately, *measuring* the NashConv of the final iteration over 1,024 random seeds is about 1,024x as expensive as measuring the NashConv of the average strategy over all iterations (and even 1,024 seeds seems insufficient to accurately measure the NashConv, so the true NashConv is likely even lower). Therefore, in order to keep the computation tractable, we measure the NashConv of the average strategy of single runs.
>
> **“On the right plot, why not include all (or at least multiple) independent plots, to give a more complete picture of the variance?”:**
> We produced a plot separated by phase (similar to Figure 4) that included the exploitability of the average of independent RM strategies. We did not include it because it is hard to interpret and did not seem particularly informative.
>
> We have updated Section 3.2 to mention the number of ERM iterations used in our experiments.

---

> > ### Comment · AnonReviewer4 · 2020-11-20
> > **Response**
> >
> > Thank you for the updates and clarifications, and apologies for the cut-and-paste error in my comments.
> > It addresses my concerns. I think there's now enough information that someone could produce similar results -- without necessarily reading through code (which does help too).
> >
> > It's a minor thing at this point, but I am still slightly unsure about exactly what is going on in figure 1, on the left side. There seems like a disconnect between sampling a specific number of actions, and the text description which controls samples using M*k_i.
> > Is the plot looking at directly sampling n different subgame actions, rather than using the M*k_i formula for sampling and varying M? Are there different runs with different values of M, and the scores are somehow aggregated over situations where the number of sampled actions corresponds to certain specific numbers of sampled actions?  (Seems unlikely, given the data.) Or is the x axis of the graph varying M, not the number of search actions directly?
> >
> > The author response is a great clarification of the different averages used in Figure 3, and why they were used. I appreciate that the level of detail here doesn't fit in the text, but I found it useful and helpful -- maybe some version of it could fit in the appendix?

---

> > > ### Author Response · Authors · 2020-11-24
> > > **Response**
> > >
> > > Figure 1 (left) is showing subgame actions on the x axis, not the parameter M. For the final version of the paper, we could also include a figure showing the performance of search with M on the x axis, and put one of the two figures in the appendix.
> > >
> > > We will either include the clarification about using the last iteration in the appendix, or include it in a separate writeup that can be cited in this paper.

---

### Official Review · AnonReviewer3 · 2020-10-26
**A thorough application of an intriguing approach to an interesting domain**

**Rating:** 7
**Confidence:** 4

**Review:**

In this paper, the authors apply an interesting twist on 1-ply search to the problem of playing no-press Diplomacy. Diplomacy is an especially interesting application, because it is neither zero-sum nor two-player, in contrast to many recent AI success stories.  Before each action, they compute an equilibrium of the next step of the game, assuming that each player will thereafter play according to a "blueprint" strategy learned via imitation learning.  The equilibrium is computed using regret matching.  The resulting agent has very strong performance against both the previous state of the art bots, and against expert human players.

The paper is clearly organized and well written; I thoroughly enjoyed reading it.  The empirical evaluations are careful.  I especially appreciated that the improved blueprint strategy was evaluated on its own (i.e., without search) to clearly separate which benefits came from the improved representation vs. the addition of 1-ply equilibrium search.

The idea of finding the equilibrium for each "stage game" of an extensive form game was new to me, and it appears to be very effective.  My main concern with this paper is that I would have liked to see some sort of justification for why we would expect this to work and where we should expect it to go wrong.  The fact that Diplomacy is non-zero-sum means that we no longer have strong theoretical guarantees about equilibrium's being the "right thing" to play. Should we expect this technique to be broadly applicable, or is it exploiting something specific to Diplomacy?

The equilibrium-search approach is in contrast with simply attempting to best respond to the blueprint strategy (apparently this is the approach of [Anthony 2020]?)  How does your technique compare to this best-response technique?  I see empirical comparisons to [Paquette 2019] but not [Anthony 2020].

I was surprised by just how computationally expensive this technique turns out to be; that is surely a drawback that will need to be addressed.

Overall I think this paper makes a valuable and well executed contribution, and I recommend acceptance.  I have some further minor comments below.

=== Minor comments ===

- p.4: "We add an additional value head...": Can you give some details about the motivation behind this architecture change?

- p.6: "If the bot's performance for each of the 7 powers is weighted equally, this score increases...": this sentence didn't make sense to me.  Possibly it assumes more familiarity with Diplomacy than I have?

- p.6: "This can partly be interpreted through an evolutionary lens...": What does the ESS view add that is not already present in the equilibrium view?  I would either flesh this out a lot more or drop references to ESS entirely.

- p.6: The definition of exploitability on p.6 is wrong because it doesn't subtract off $u_i(\pi)$ (also inconsistent with definition on p.8, which appears to be correct)

---

> ### Author Response · Authors · 2020-11-20
> **Response**
>
> Thank you for your review!
>
> In order to aid reproducibility, we will open source the code for training and running our agent.
>
> **“When should we expect this to work?”:**
> That's a great question and we're interested in exploring it in future work, but we were hesitant to speculate in the paper without justification. Our primary goal is to provide a clear and convincing datapoint for when regret minimization can be successful in a complex game. But for the purposes of speculation, here are some thoughts:
>
> - ERM/CFR converges to a Nash equilibrium in games other than just two-player zero-sum games. For example, it is proven to converge to a Nash equilibrium in zero-sum polymatrix games (a generalization of two-player zero-sum games to multiple players; see [Cai et al., Mathematics of Operations Research 2016] which proves that coarse correlated equilibria are equivalent to Nash equilibria in zero-sum polymatrix games, and note that ERM produces a coarse correlated equilibrium). Diplomacy may be close to a class of games in which ERM can approximate a Nash equilibrium. However, we have also observed many situations in Diplomacy where ERM fails to converge to a Nash equilibrium. In these situations, it may be that ERM resembles the reasoning process of humans, and is able to find solutions that are compatible with their strategies.
>
> - Even if ERM converges to a Nash equilibrium, it could converge to the “wrong” equilibrium (i.e., one that performs poorly in a human game). By assuming the blueprint policy (which is based on human data) is played by all players for future phases, we hopefully are more likely to approximate an equilibrium that is consistent with human play. For example, if our agent considers “backstabbing” an ally, the agent will recognize that next turn the ally will likely retaliate (e.g., tit-for-tat).
>
> - Another major component of this work is showing how effective search can be. Search has already been extremely successful in prior games like backgammon, chess, go, poker, and hanabi. We believe this work reinforces the point that search is extremely powerful, and especially that searching for *all* players (rather than simply computing a best response for one player) is very valuable. A major limitation of search is that it requires a simulation of the environment. This is available in games like Go and Diplomacy, but not in most real-world interactions. Nevertheless, recent work like MuZero (Schrittwieser et al. 2019) increasingly makes it possible to deploy search even without a simulator.
>
> **Regarding a comparison to simply computing a best response to the blueprint in the “stage” (which we will call BRBot) rather than an equilibrium (SearchBot):**
> We have run this experiment. 1 BRBot vs. 6 SearchBots scores 11.1%, while 1 SearchBot vs. 6 BRBots scores 17.2%. As a reminder, a tie would be 14.3%. BRBot does better than SearchBot when playing against 6 blueprint agents (as expected, since it is best responding to the blueprint agent). We will add a table of BRBot playing against each agent to the final version of the paper. While we have not measured BRBot’s exploitability, we believe that SearchBot is almost certainly less exploitable than BRBot.
>
> While one would expect SearchBot to have lower exploitability than BRBot, the better head-to-head performance might appear surprising. Our intuition is that BRBot may overfit to the weaknesses of the blueprint: it may choose actions that exploit the specific mistakes of the blueprint, but those actions might in general be suboptimal. By computing an equilibrium, SearchBot may be more likely to avoid these suboptimal actions.
>
> **Regarding a comparison to Anthony et al.:**
> Their agent is not open-sourced, but we will compare SearchBot to their agent once it is available. Anthony et al. do not just compute a best response to the blueprint in the current stage (as BRBot does). Instead they train in an iterative fashion where the blueprint changes as well.
>
> **“Why do you add an additional value head?”:**
> We use this value estimate during equilibrium search to estimate the value of a Monte Carlo rollout after a fixed number of steps. We updated the text to reflect that.
>
> **Regarding "If the bot's performance for each of the 7 powers is weighted equally, this score increases...":**
> We expanded the explanation for this, but moved it to a footnote in order to not distract from the main point of the section.
>
> **“What does the ESS view add that is not already present in the equilibrium view?”:**
> This is a good point. We have removed the ESS paragraph, though if other reviewers found it helpful then we are happy to add it back in.
>
> We’ve corrected the exploitability definition on p. 6. Thank you for pointing it out.

---

### Official Review · AnonReviewer2 · 2020-11-02
**Human-Level Performance in No-Press Diplomacy**

**Rating:** 8
**Confidence:** 4

**Review:**

The authors consider "no-press Diplomacy", a complex game played by humans which involves (limited) cooperation and competition.

The method uses a policy and value function learned from human games, together with a test-time search. The imitation-learned policy is used both to restrict the actions considered in the search and also to roll out the leaf nodes of the search (to a fixed depth, after which the value function is used). The search process is a sampled form of external regret matching.

Several wrinkles required for good performance are clearly motivated and explained, e.g. handling low-entropy policies and large action spaces.

The authors evaluate against existing bots, against multiple human players on webdiplomacy.net, and against two human experts. In all cases, the authors algorithm outperforms its opponents.

It is not feasible to compute exact exploitability in a game of this size, or even to train an approximate best response against the searching agent. The authors therefore train an approximate best-response RL agent against both the imitation-learned policy and a distillation of the searchbot, strongly suggesting that the distilled search is less exploitable than the imitation-learned policy.

The method described is novel, but also a fairly straightforward extension of prior work to this domain, incorporating a single-ply search on top of an imitation-learned policy. Although there is nothing radically new, the combination of very strong empirical results, and clear & detailed explanation of the methods involved combine to make this a clear accept. I was particularly happy to see the thorough evaluations including bots, a field of humans, and world-class players, and an attempt at investigating exploitability.

Comments on the paper:
* In figure 3, it might be clearer to plot the two graphs on the same scale.
* In the Qualitative Assessment of SearchBot section, I would have been interested in any comments the human experts might have made.

---

> ### Author Response · Authors · 2020-11-20
> **Response**
>
> Thank you for your feedback!
>
> In order to aid reproducibility, we will open source the code for training and running our agent.
>
> We will plot both graphs in Figure 3 on the same scale in the final version of the paper.
>
> Regarding a qualitative assessment of the bot from human experts, we will ask an expert gunboat player to review the bot’s games on webdiplomacy.net and provide comments.

---

### Official Review · AnonReviewer5 · 2020-11-05
**First demonstration of human-level performance on an important domain**

**Rating:** 7
**Confidence:** 5

**Review:**

# Summary

The paper addresses the problem of No Press Diplomacy. It applies existing search techniques to improve on an imitation policy, which is learned with improvements on existing supervised learning methods for the domain. The resulting policy is tested against previous learned agents and humans, showing impressive results.

I recommend accepting this paper. It tackles an important multiagent environment, producing the first agent to demonstrate human-level play. While the techniques used are not especially novel, their success in a complex mixed-motive 7-player game is interesting. I have no major concerns about the paper, except for the specific claim that the agent 'convincingly surpasses human performance' - I think this is overstated and should be removed before acceptance.

 # Strengths

Diplomacy is an important multiagent domain, involving a complex action space and a mixture of cooperation and competition between players, which necessitates new approaches. Research on this domain is timely, building on recent work as well as a long history of AI research. This paper advances the state of the art in this domain.

The search method in the paper (external regret matching on a subsampled game, evaluated with on-policy rollouts) is clearly explained and situated in the existing literature. While it is not particularly novel, the successful application to this type of domain is.

Overall, the experiments performed in the paper demonstrate convincingly that SearchBot is a large improvement on policies learned from imitation, and that it is human level.
 * The results in ad-hoc play with humans are impressive. They clearly show that SearchBot is the standard of a fairly strong human player. This is a particularly important contribution in the domain of Diplomacy; I agree with the authors that performance against humans is the ultimate test, and in a 7-player game it is not clear a priori that an agent that fares well against other agents will also do well against humans.

* Play with expert humans shows that SearchBot is not overly exploitable; it is impressive that experts get well below the average result in this setting.

* The agent exploitability analysis is good, showing that SearchBot is (almost certainly) less exploitable by RL than the blueprint it is based on.

The paper is well-written; related work is addressed well, and the methods and findings are for the most part very clear.

# Weaknesses

## Experiments

I think the claim made in the introduction that it 'convincingly surpasses human-level performance' is too strong, and should be removed or clarified. The natural reading of this is that SearchBot is better than the best human players; but the results in the paper do not bear this out. The reported ranking is 23/1128, and many players in this ranking have played very few games and are unlikely to have an accurate rating. The descriptions elsewhere in the paper (such as the title and conclusions) of 'human-level performance' seem to me to be the appropriate strength of claim.

It is unclear how strong the human players SearchBot faced in the webdiplomacy evaluation were. I think it would be very useful to include some information on this, and on how SearchBot fares against players of different skill levels. This may well affect how meaningful the reported rank is; if SearchBot has overwhelmingly played rather weak players, its rank is probably not comparable with top human players, who presumably face much stronger opposition.

The paper compares to two other agents; the SL and RL version of DipNet. Previous work has compared to the rules-based agent Albert; this has the advantage that this bot is independent of human imitation data. Adding this comparison would be valuable, and would help substantiate the strong claim that SearchBot ‘greatly exceeds the performance of past no-press Diplomacy bots’.

The effects of two key search parameters - rollout length and subgame size - are investigated. However, the effect of the number of iterations of ERM on play strength is not addressed in the paper; I would like to see a similar experiment to those in Figure 1 which looks at this.

Some winrates are presented with confidence intervals, but some in Tables 3, 5 and 6 are not. These should be added.

## Reproducibility

The methods and results are generally well explained, but I think there are a few places where there could be more detail:
* The experiments which do not involve humans do not have a clear explanation of the hyperparameters used (particularly for search). A section like Appendix F should be added for this.
* The experiments between bots used the SoS rules. However, I could not find a description of the circumstances under which games were declared draws.
* The imitation learning architecture is described only as changes to previous work; I think setting out the entire architecture in an appendix would aid reproducibility, as currently the reader needs to read two other papers to reproduce this.

---

> ### Author Response · Authors · 2020-11-20
> **Author Response**
>
> Thank you for your feedback!
>
> In order to aid reproducibility, we will open source the code for training and running our agent.
>
> We agree that the agent demonstrates human-level performance but not superhuman performance. We have changed the wording in the introduction to “achieves human-level performance” to clarify this point.
>
> Regarding the strength of the players that SearchBot played against on webdiplomacy, there is certainly a mix of strength levels. For this reason we also reported the Ghost-Ratings score of the bot, which accounts for the strength of opponents in much the same way that Elo rating accounts for the strength of opponents in games like chess and Go. The bot’s ranking of 23rd is based on the Ghost-Rating.
>
> Regarding a comparison to Albert, we have run 25 1v6 games against Albert and SearchBot scored 69.3% according to sum-of-squares scoring. The games ended in a draw after 1935 or if no power gained a center for 2 years. We acknowledge this is a small number of games. We plan to run more games against Albert and also evaluate against the agent of Anthony et al. once the agent is available for testing. We will include these results in the final version of the paper.
>
> “Effect of #ERM iterations on head-to-head performance in Fig. 1”:
> This is a great suggestion, we added this panel to Figure 1.
>
> Regarding confidence intervals on tables 3 and 6:
> We believe confidence intervals would not be appropriate because the samples are not independent. Indeed, the intention was that the humans would adapt and improve over time. For table 5 we added the standard error for individual powers.
>
> The hyperparams used in bot games were the same as used in live human games. We have updated the text to say this.
>
> We have added information to Section F describing draw criteria for human and bot games.
>
> We have included a complete description of the imitation learning architecture in Appendix A.

---

> > ### Comment · AnonReviewer5 · 2020-11-20
> > **Response**
> >
> > Thank you, this addresses my concerns well. The addition to Figure 1 is valuable, and the comparison against Albert is already useful in that it clearly exceeds the performance of DipNet (though more games will obviously be better). The clarifying updates are good, and I agree with the decision on confidence intervals in tables 3 and 6.
> >
> > I still wonder is whether the Ghost Ranking of 23rd is a true reflection of the bot's strength. While this is a good metric to report, the bot's number of games and opponent distribution is probably very atypical for strong players, and I do not have confidence that Ghost Rating is accurate in that case. Still, this is a minor point - the key claim of 'human-level performance' is well supported.

---

### Decision · Program_Chairs · 2021-01-07
**Final Decision**

**Decision:**

Accept (Oral)

**Comment:**

All reviewers agree that this paper is very solid work, that presents great progress in no-press diplomacy. The method and presented experiments are of very good quality and the work merits to be presented at ICLR.